# Photoacoustic 2D actuator via femtosecond pulsed laser action on van der Waals interfaces

Xin Chen [1,2], Ivan M. Kislyakov [1] ✉, Tiejun Wang [3], Yafeng Xie[1,2], Yan Wang[1,2], Long Zhang[1,4] & Jun Wang [1,2,3,4] ✉

Achieving optically controlled nanomachine engineering can satisfy the touch-free and non-invasive demands of optoelectronics, nanotechnology, and biology. Traditional optical manipulations are mainly based on optical and photophoresis forces, and they usually drive particles in gas or liquid environments. However, the development of an optical drive in a non-fluidic environment, such as on a strong van der Waals interface, remains difficult. Herein, we describe an efficient 2D nanosheet actuator directed by an orthogonal femtosecond laser, where 2D $VSe_2$ and $TiSe_2$ nanosheets deposited on sapphire substrates can overcome the interface van der Waals forces (tens and hundreds of megapascals of surface density) and move on the horizontal surfaces. We attribute the observed optical actuation to the momentum generated by the laser-induced asymmetric thermal stress and surface acoustic waves inside the nanosheets. 2D semimetals with high absorption coefficient can enrich the family of materials suitable to implement optically controlled nanomachines on flat surfaces.

The idea that light carries forces to manipulate particles dates back to Kepler and Newton and was later confirmed by Maxwell[1]. Since the pioneering work by A. Ashkin and F. Ehrenhaft, optical and photophoretic forces are the most common methods of optical manipulation[2,3]. The optical force, originating from momentum conservation and ponderomotive action of light, is usually small and mainly utilized to transport three-dimensional (3D)[4–6], or suspended two-dimensional (2D)[7] particles. Meanwhile, the photophoretic force, generated by laser-induced nonuniform heating of surrounding gas around the particle, can also effectively control the motion of particles[8,9]. Optical manipulation has been gradually optimized using structured beams, structured particles, and structured background[10–15] and has realized multi-dimensional control such as push, pull, lateral movement, and rotation, as well as large-size particle transfer and long-distance transportation[9,15–19]. Benefiting from the advantage of non-touch and non-invasive properties, optical manipulation has the potential for applications in nanotechnology, biology, and mechanics[8,15,20,21]. However, these light-induced forces work mostly in vacuum, ambient air, and liquids to decrease environmental resistance. The latest related researches report the laser-driven complex motion of gold nanoplates on fixed microfibers, where photoacoustic Lamb waves overcome friction at the plane-cylinder contact line and make the nanoplates rotate around the fiber and slide along it[22–24]. However, the contact area between nanoplates and microfibers is relatively small, utilizing optical manipulation to achieve 2D planar motion is still not observed. Achieving the optical drive for particles in strong van der Waals (vdW) interactions is still a big challenge, which needs to be addressed in practical applications.

[1]Photonic Integrated Circuits Center, Key Laboratory of Materials for High-Power Laser, Shanghai Institute of Optics and Fine Mechanics, Chinese Academy of Sciences, Shanghai 201800, China. [2]Center of Materials Science and Optoelectronics Engineering, University of Chinese Academy of Sciences, Beijing 100049, China. [3]State Key Laboratory of High Field Laser Physics, Shanghai Institute of Optics and Fine Mechanics, Chinese Academy of Sciences, Shanghai 201800, China. [4]Center for Excellence in Ultra-intense Laser Science, Chinese Academy of Sciences, Shanghai 201800, China. ✉e-mail: iv.kis@siom.ac.cn; jwang@siom.ac.cn

Herein, we report an intriguing performance of 2D material behavior under ultrafast laser irradiation, where the VSe$_2$ and TiSe$_2$ nanosheets in a dry contact with flat sapphire and quartz glass substrates were actuated by femtosecond (fs) pulsed laser, indicating that large vdW interactions between nanosheets and substrates were overcome. Furthermore, we investigate the properties of various 2D materials to find the sensitive parameters in the optical drive. We analyze its possible mechanisms in terms of the magnitudes of the forces acting in them and present our understanding of the nature of the discovered phenomenon based on a photoacoustic mechanism. Air gaps between nanosheet edges and the substrate observed via cross-sectional transmission electron microscopy (TEM) inevitably lead to local overheating, which causes asymmetric thermal stress and surface acoustic waves. Through the analysis of different 2D material-substrate systems, a large absorption coefficient (~ $10^5$ cm$^{-1}$) seems to be the crucial property for this optical drive. For such materials, the thermal expansion coefficient and the bulk modulus also become important. The maximal speed efficiency observed in the VSe$_2$ actuator is 434 µm·s$^{-1}$·mW$^{-1}$ in our experiment, which is much higher than efficiencies of previous reported photoacoustic manipulations. Therefore, we present a method to overcome the adhesion forces of 2D materials with substrates, which expands application possibilities of 2D materials as nanomachines.

## Results and discussions
### Optical drive

A VSe$_2$ nanosheet was attached on a horizontal polished sapphire substrate via mechanical exfoliation (ME) method, and it was completely covered by a vertical pulsed laser beam, as shown in Fig. 1a (See details in Methods section). When the pulsed laser is applied, the VSe$_2$ nanosheet begins to move, and continues moving in the region of uniform illumination. Then moving the light source in the direction towards the nanosheet provides the uniformity of its motion. The character of motion is shown in Fig. 1b and Supplementary Movie 1. Due to the connection to the substrate, the motion of the nanosheet is limited to the substrate surface plane ($x$, $y$-plane). This light-induced movements indicate that an optical-to-mechanical energy conversion occurs in this actuator system.

In our experiment, the pulse mode of irradiation is a crucial factor for driving the nanosheets. When we switch the driving source to a continuous-wave (CW) laser (the wavelength $\lambda = 1064$ nm) or to a high repetition frequency laser (the repetition frequency $f_{rep} = 80$ MHz, $\lambda = 800$ nm), we do not observe the motion of the nanosheet. Additionally, there was no obvious correlation in the speed of their motion with geometric dimensions and shape. For example, Supplementary Movie 2 shows the movements of a triangular VSe$_2$ nanosheet, which implies that the motion behavior is not restricted by the shape of the nanosheets. Besides, the Raman spectra of a nanosheet before and after laser irradiation (See details in Supplementary Note 2) illustrates that the nanosheets are indeed VSe$_2$ with the 1 T phase[25]. The Raman peaks remained unchanged after laser irradiation, indicating that the possibility of laser-induced chemical transformations and oxidation can be excluded in our experiment[26]. Notably, the actuator system has good durability against prolonged pulsed laser exposure.

To further evaluate the effect of motion, we determined the speed of a VSe$_2$ nanosheet. Figure 2a shows the traveling speed as a function of the repetition frequency, at a fixed pulse energy. It shows a linear growth of the speed with the repetition frequency, which suggests that each laser pulse can actuate the nanosheet to move a certain distance (Fig. 2b), and the movement induced by each pulse is independent of each other. Figure 2c shows the relationship between the speed and the laser pulse energy. It increases linearly with laser energy (Fig. 2d) and especially has a step-like threshold. We associate this behavior with the manifestation of the dual character of adhesive friction forces consisting of static and sliding friction[27]. When the laser energy is small, it is all spent on overcoming the static friction force, and when the energy noticeably exceeds this threshold force, then most of it is spent on overcoming the sliding friction force. Thus, by measuring speed-energy characteristic, we can test the static/sliding ratio of the

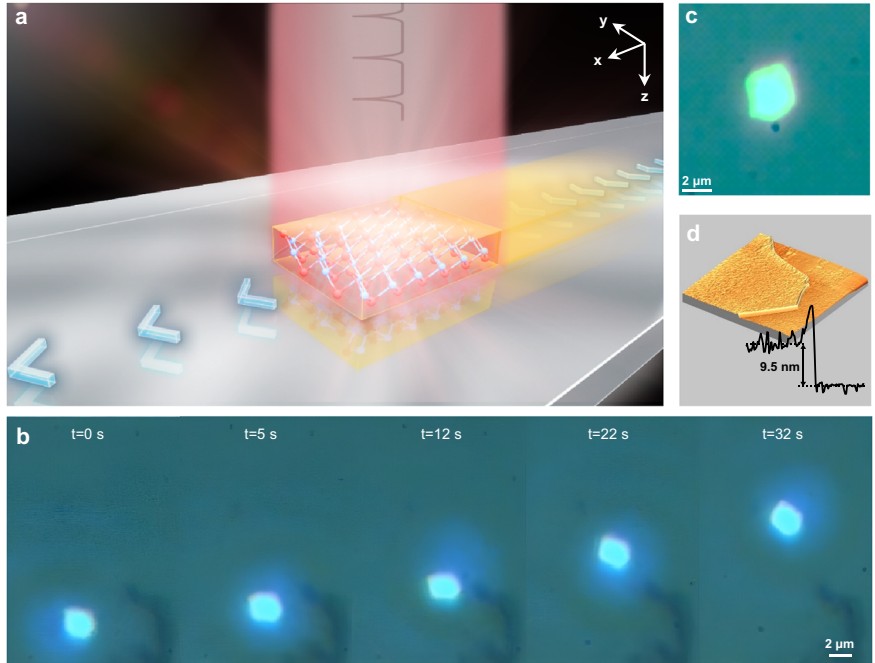

**Fig. 1 | Optical drive for 2D materials nanosheets on sapphire substrate.**
**a** Schematic of 2D nanosheet actuator on polished sapphire substrate. The laser direction is along $z$-axis, and the nanosheet moves on the $x$,$y$-plane. **b** Sequencing images of a VSe$_2$ nanosheet (3.7 µm × 3.1 µm) moving on the sapphire substrate under the manipulation of the near-flat fs pulsed laser (See Supplementary Movie 1). The laser was at 1040 nm with the energy and the repetition frequency of approximately 13 nJ and 1 kHz. **c** Optical image of the VSe$_2$ nanosheet. **d** 3D atomic force microscopy characterization of VSe$_2$ nanosheet.

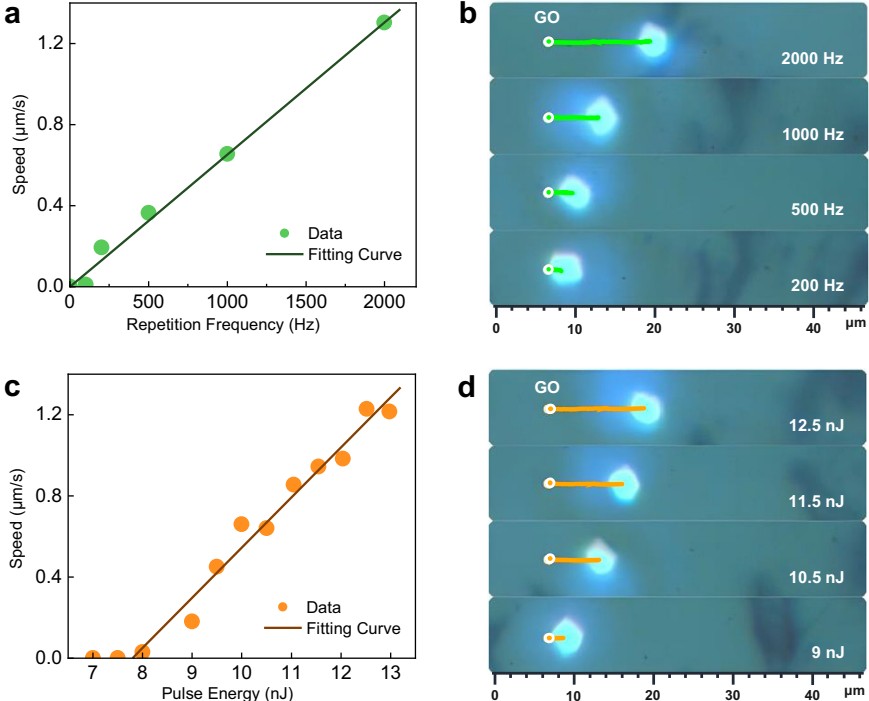

**Fig. 2 | Speed of the VSe$_2$ nanosheet actuator. a** Relationship between light-actuated moving speed of the nanosheet at $E_p = 10$ nJ and the laser repetition frequency: experimental points and their linear fit. **b** Corresponding moving curves (green lines) in nearly 10 s at 200 Hz, 500 Hz, 1 kHz, and 2 kHz. The moving distance increases with the increasing repetition frequency. **c** Relationship between light-actuated moving speed of the nanosheet at $f_{rep} = 1$ kHz and the laser pulse energy: experimental points and their linear fit. **d** Corresponding trajectory curves (orange lines) in nearly 10 s at 9 nJ, 10.5 nJ, 11.5 nJ, and 12.5 nJ. The moving distance increases with the increasing laser energy.

vdW friction force. The static friction force corresponds to the step position (i.e., 9 nJ in Fig. 2c), while the sliding friction force corresponds to the *E*-intercept of the linear approximation (7.8 nJ).

## Van der Waals adhesion and friction forces

The VSe$_2$ nanosheets were tightly attached to the sapphire substrate via ME method. The atomic force microscope (AFM) image shows the roughness of the substrate is less than 0.2 nm (See details in Supplementary Note 3), which guarantees a strong dry adhesion force at the interface between the nanosheet and the substrate. Therefore, the vdW force is dominated between them[28]. Assuming that the nanosheet and the substrate are two parallel smooth surfaces, the vdW force $F_{vdW}$ can be calculated by

$$F_{vdW} = -\frac{AS}{6\pi d^3} \quad (1)$$

where $A$ is the Hamaker constant, $S$ and $d$ are the contact area and separation distance between the two surfaces, respectively[27]. The constant $A$ is tightly related to the permittivity of the involved materials. For metals on dielectric substrates, the value can be roughly estimated as:[29]

$$A = \frac{3h}{8\sqrt{2}} \frac{(n_s^2 - 1)v_p v_s}{\sqrt{(n_s^2 + 1)}\left[\sqrt{(n_s^2 + 1)} v_s + v_p\right]} \quad (2)$$

where $h$ is the Planck constant, $n_s$ is the refractive index of substrate at high frequencies (for sapphire $n_s = 1.75$)[27], $v_s$ is the main electronic absorption frequency of sapphire ($v_s = 3.2 \times 10^{15}$ s$^{-1}$, $hv_s = 13.23$ eV)[27], and $v_p$ is the plasma frequency of the metal. The averaging over crystal orientations gives $h\bar{v}_p = 3.23$ eV for VSe$_2$[30]. The resulting Hamaker constant is $A = 0.39$ eV.

The separation distance $d$ can be evaluated via a rigid sphere model as the distance between centers of the lowest spheres of the top surface and the highest spheres of the bottom, when the hard sphere radii of the surface atoms (oxygen from the sapphire side and selenium from VSe$_2$ side) are taken as half minimal O · O and Se · Se distances in the contacting planes, respectively: $r_O = 0.126$ nm[31] and $r_{Se} = a_{lat}/2 = 0.168$ nm[32] (where $a_{lat}$ is the crystal lattice parameter). Oxygen atoms in the contact surface of *c*-sapphire form a face of hexagonal close-packed lattice[33], and with a tight package, atoms of the top surface should lie in hollows of equilateral triangles formed by oxygen atoms[29]. Then it is easy to find $d = \sqrt{(r_O + r_{Se})^2 - 4r_O^2/3} = 0.255$ nm.

To test our calculations, we cut one of the nanosheets together with the sapphire substrate by a focused ion beam (FIB) and examined the contact area with a high-resolution TEM, as shown in Fig. 3 (See details in Supplementary Note 4). Figure 3d indicates the interlayer distance of $0.224 \pm 0.034$ nm in the central part of the nanosheet. It is in agreement with similar observation of the VSe$_2$-sapphire contact[33], and the value by hard-sphere model corresponds to these both observations. Based on the latter value and using the above $A$ value, the surface density of the vdW force between the VSe$_2$ nanosheet and sapphire substrate ("vdW pressure") calculated by Eq. (1) holds $F_{vdW}/S = 0.200$ GPa.

It can be clearly seen in Fig. 3b, c that air gaps exist along the edges of the nanosheet. Figure 3e, f shows the enlarged images of air gaps, which are obviously caused by fractures of the layered material during its coating, so the area of close contact is always less than or equal to the visible area of the nanosheet. Therefore, the absolute value of the vdW force, determined by Eq. (1) for a specific nanosheet, is its upper estimate.

The morphology of the VSe$_2$ nanosheet, whose speed of motion was measured, is shown in Fig. 1c. Its lateral shape was approximately rectangular of the size $S = 3.7 \times 3.1$ μm$^2$. With this area, the maximal

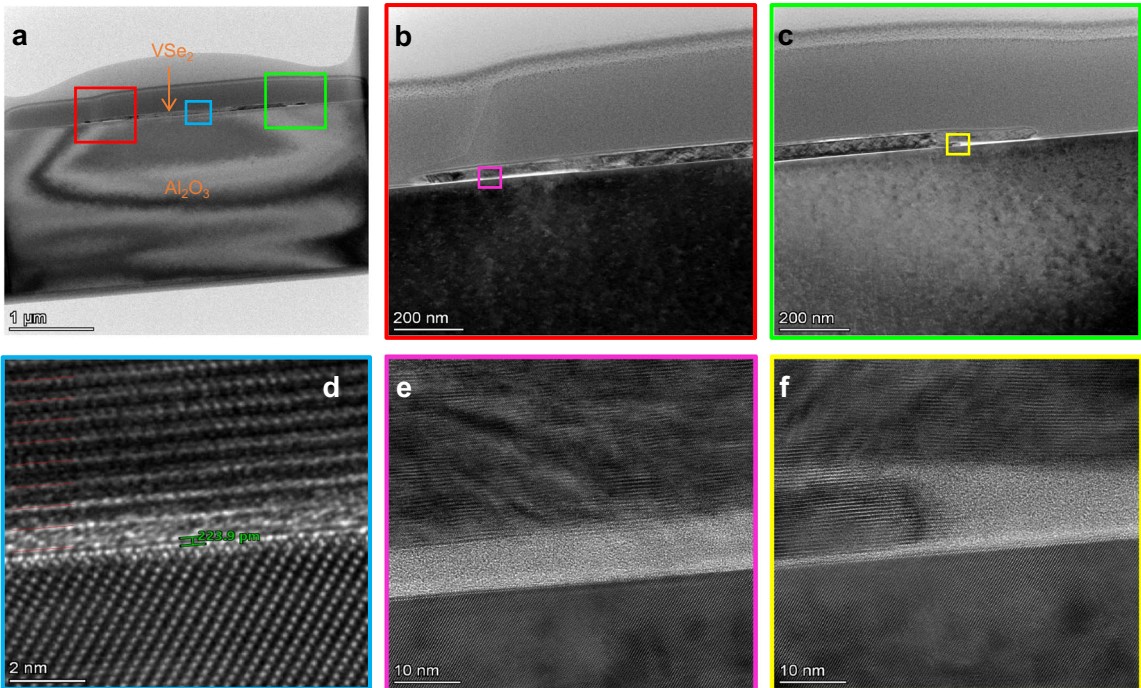

**Fig. 3 | Cross-sectional transmission electron microscopy images of a VSe$_2$ nanosheet on sapphire substrate. a** General view of the VSe$_2$-sapphire contact. The scale bar is 1 µm. **b, c** Enlarged images of the red square and green square in **a**. The scale bar is 200 nm. **d** Enlarged image of the blue square in **a**. The interlayer distance is about 0.224 nm. The scale bar is 2 nm. **e, f** Enlarged images of the pink square in **b** and the yellow square in **c**. The scale bar is 10 nm.

$F_{\text{vdW}} \approx -2.3$ mN. The thickness of the nanosheet in Fig. 1d was identified as $L = 9.5$ nm. The nanosheet gravity $\rho g S L = 6.2$ fN, i.e., negligibly small. That is, to manipulate the attached nanosheet, the laser-induced force must overcome the vdW adhesion force or, at least, the adhesion-controlled friction force (in the case of the pure horizontal sliding)[23].

The friction force can be estimated in the rigid sphere model as:[27]

$$F_{fr} = 2\varepsilon S \Delta\gamma / \delta \tag{3}$$

where $\Delta\gamma$ is the part of the adhesion surface energy consumed when the spheres of the top surface detach from those of the bottom surface in order to go over them, $\delta$ is the horizontal displacement of a sphere in one cycle of this movement, and $\varepsilon$ is the part of the kinetic energy of the moving sphere transferred to the bottom surface. Generally, $\Delta\gamma = \Delta W_{adh}/2$, the half of adhesion work spent by a sphere of the upper surface to move between the states with lowest and highest energies. The maximal energy difference will be achieved when selenium atom moves through the neighboring triangle along its mean line. At that, the atom passes two loading-unloading adhesion cycles, it is easy to obtain: $\Delta\gamma - 0.356 W_{adh}/2$ and $\delta = 2\sqrt{3}r_O$ (See details in Supplementary Note 5).

For an atom of the top surface contacting three atoms of oxygen in sapphire, an upper bound approximation is known: $W_{adh}^+ \sqrt{3}A/4\pi^2 r_O^2$ [29]. Therefore, for an ideal contact when all selenium atoms lie in the equilateral triangles of oxygen, the maximal ($\varepsilon = 1$) shear stress due to friction for VSe$_2$ will be:

$$\sigma_{fr} = \frac{F_{fr}}{S} = \frac{C_{\Delta\gamma}A}{8\pi^2 r_O^3} \tag{4}$$

where $C_{\Delta\gamma} = -0.356$. With the Hamaker constant determined above, we get $\sigma_{fr} = -0.141$ GPa. For the VSe$_2$ nanosheet with $S = 3.7 \times 3.1$ µm$^2$, the maximal friction force will be $F_{fr} = -1.62$ mN.

With thermal film growth methods such as molecular beam epitaxy, the VSe$_2$ lattice is deformed due to a large coefficient of thermal expansion, ensuring optimal location of selenium atoms onto the interstices of oxygen atoms in sapphire[33]. However, this does not happen during cold mechanical deposition, and there is a large degree of mismatch between the VSe$_2$ and sapphire lattices. This should lead to larger separation distances in the ME samples and, accordingly, weaken both the vdW force and the adhesion-controlled friction. Nevertheless, the above upper-bound estimates are important for understanding the scale of the forces in action.

We changed different substrates to create different values of vdW force, including sapphire, quartz, and silicon. The movement results are summarized in Table 1. We can see the motion efficiency ranking: quartz > sapphire > silicon. Similarly, the pressure values of vdW forces were calculated by Eq. (1) and listed in Table 1. It reveals a link between vdW forces and speed efficiency: the smaller the vdW adhesion force, the greater the efficiency of the movement. Moreover, the maximal speed efficiency that we observed in our experiment is 434 µm·s$^{-1}$ per milliwatt, which is several times higher than the efficiency values shown in both photophoretic[9] and photoacoustic[22,24] manipulations.

## Optical forces

Many different mechanisms of the propelling effect of light on material particles are known. An important group of them concerns direct light-matter interactions, which is called optical force and consists in transmission of the mechanical momentum of light to matter[2]. It can be due to both scattering and gradient forces. The scattering force uplifts cambered nanoparticles in the focused beams due to refraction

**Table 1 | Manifestation of the optical driving in different substrates**

| Nanosheet-Substrate System | Speed (µm·s$^{-1}$)/ Energy (nJ) | Efficiency of Movement (µm·s$^{-1}$·mW$^{-1}$) | vdW Pressure (GPa) |
|---|---|---|---|
| VSe$_2$-Quartz | 2.78 / 6.4 | 434 | 0.044 |
| VSe$_2$-Sapphire | 1.83 / 5.0 | 366 | 0.200 |
| VSe$_2$-Silicon | Motionless | 0 | 0.211 |

at the oblique incidence of the beam on their curved surfaces[6,34]. In the case of a flat surface light can transfer momentum only toward the substrate, and the force value $-P/c = 0.1$ mN is too small at the laser pulse power $P$ in our experiment. Therefore, we exclude the participation of this part of the optical force in the observed motion.

The transversal forces exerting on a nanoparticle from light come from phase and intensity gradients[35]. Despite the piconewton values of the phase gradient forces, their application for sorting metallic nanoparticles has been reported[36]. The phase gradient in the focus of a Gaussian beam is zero, and an appropriate phase mask is necessary to generate the forces. Although the apodizing filter that we use in our experiment is radially symmetric, alignment errors can bring to the appearance of some phase gradient. However, our careful observations of the nanosheet movement with the transversal shifting of the filter indicate the absence of the influence of these forces.

The intensity gradient forces are ponderomotive[37], and in our case originates from the gradient of the light intensity, $\nabla I$, in the surface of the nanosheet: $F_{grad} = 2V(\varepsilon - 1)(\varepsilon + 2)\nabla I/3c$ (See details in Supplementary Note 6). Directed towards the increment of the intensity gradient, this force may influence the lateral deviation of the particle and fix it at the center of the light focus[5]. With an apodizing filter, the radial gradient was substantially reduced in our experiments, except the edge intensity gradients, which did not surpass $1 \times 10^{21}$ W/m$^3$ (assuming the width of the spot boundary of the order of wavelength: $dx$ ~1 μm). In a nanosheet with a characteristic length $a = 3.7$ μm, $\nabla I \neq 0$ only within a streak of the width $a_x = 1$ μm. Therefore, the volume affected by the force is $V = aa_xL$. With an experimental value of permittivity of VSe$_2$: $\varepsilon(1040 \text{ nm}) = 1.6$[38], we get the assessment $F_{grad} = 178$ nN. This small value, as well as the observed behavior of the nanosheet, on which the force acts mainly in the center of the light spot and weakens towards the periphery, allow us to conclude that optical forces do not participate in the observed effect.

## Photoacoustic mechanisms

The pulsed laser was a crucial factor for driving the nanosheet in our experiment. In 1963, R. M. White considered an instantaneous thermal excitation caused by a pulsed laser on the metal surface[39]. The lattice units can expand and shrink under the irritation of the pulsed laser, and eventually elastic acoustic waves are formed. When the nanosheet thickness is much smaller than the sound wavelength (as in our case), two acoustic Lamb waves (symmetric and antisymmetric) can arise and propagate over the surface of the nanosheet[40]. These waves induce translational displacement of the material, and thus can move the nanosheet when its connection to the substrate is weakened[22–24].

The occurrence of photoacoustic force originates from four main mechanisms: piezoelectric, electrostrictive, electronic (through the deformation potential), and thermoelastic[41,42]. We do not consider the inverse piezoelectric and electrostrictive effects. The former is because no piezoelectric properties can be expected due to inversion symmetry in the 1T-VSe$_2$[43]. The latter is because the nanosheet thickness is much smaller than the wavelength, and the field gradient in $z$ direction required to drive electrostriction is negligible. The electronic mechanism of lattice deformation due to the detachment of bound electrons can also be ignored, because our excitation energy (1.19 eV) is sufficiently lower than the interband excitation energy (1.6 eV in VSe$_2$)[44]. Therefore, we only focused on the thermoelastic mechanism of the sound excitation.

We studied several pairs of materials in the nanosheet-substrate system. Different 2D materials, including h-BN, MoS$_2$, WSe$_2$, PdSe$_2$, TiSe$_2$, and VSe$_2$, were exfoliated via the same ME method and deposited on the same sapphire substrates. We also prepared a VSe$_2$-sapphire system via chemical vapor deposition (CVD) method. Their movement results are summarized in Table 2. We can see that our optical drive system can only actuate TiSe$_2$ and VSe$_2$ (both ME- and CVD-VSe$_2$) nanosheets, it is powerless to actuate h-BN, MoS$_2$, WSe$_2$,

**Table 2 | Manifestation of the optical driving in different nanosheet/substrates systems**

| Nanosheet-Substrate System | Movement Results | Speed (μm/s)/Energy (nJ) | Efficiency of Movement (μm·s$^{-1}$·mW$^{-1}$) |
|---|---|---|---|
| h-BN-Sapphire | No | - | - |
| MoS$_2$-Sapphire | No | - | - |
| WSe$_2$-Sapphire | No | - | - |
| PdSe$_2$-Sapphire | No | - | - |
| TiSe$_2$-Sapphire | Yes | 0.70/11.0 | 64 |
| VSe$_2$-Sapphire | Yes | 1.83/5.0 | 366 |
| VSe$_2$(CVD)-Sapphire | Yes | 1.15/8.4 | 136 |

and PdSe$_2$ on the sapphire substrates. This phenomenon drives us to analyze which parameters of materials dominate the movements.

A number of physical properties of the studied materials that may be related to this mechanism has been collected and compared, such as absorption coefficient $\alpha$, reflectivity $R$, linear thermal expansion coefficient $\alpha_{TE}$, thermal conductivity $\kappa$, specific heat $C_p$, density $\rho$, average (upper bounds polycrystals) values of bulk modulus $B$, Young's modulus $E$, and Poisson's ratio $v$ (See details in Supplementary Note 7). It can be seen that VSe$_2$ and TiSe$_2$ stand out for their huge values of the absorption coefficient, which are at least one order of magnitude higher than that of other 2D materials. Since there is a discrepancy between the reported experimental and theoretical data on the absorption coefficients, we took an average value $\alpha_{VSe_2} = 5.6 \times 10^5$ cm$^{-1}$ for the following calculations.

The second significant difference of VSe$_2$ and TiSe$_2$ from other materials is their greater values of the linear thermal expansion coefficient in the layer plane. These features indicate rather the thermoelastic nature of the nanosheet movement, which can manifest itself under the condition of inhomogeneous heating. In this case, the nanosheet can acquire a temperature gradient when irradiated, leading to an in-plane thermal stress, which determines non-uniform stretching and thus acoustic waves in the nanosheet plane.

The heating of an absorbing uniform film by light can be found by solving the one-dimensional heat equation:

$$\frac{\partial T(z,t)}{\partial t} = \chi \frac{\partial^2 T(z,t)}{\partial z^2} + q_s(z,t) \qquad (5)$$

where $z$ is the beam propagation coordinate, $\chi = \kappa/\rho C_p$ is the heat diffusivity, and $q_s(z,t)$ is the heating source function. Assuming that the intensity distribution of the laser beam is radially symmetric, the Lambert-Beer absorption in the case of normal incidence gives:[41]

$$q_s(z,t) = -\frac{(\nabla \cdot S)}{\rho C_p} = \frac{\alpha(1-R)E_p e^{-\alpha z}}{\pi \omega_0^2 \rho C_p} \frac{f(t)}{\int_0^\infty f(t)dt} \qquad (6)$$

Here $S$ is the Poynting vector, $R$ is the reflectivity at the moment of absorption (stationary value), $E_p$ is the pulse energy, $w_0$ is the laser spot radius, and $f(t)$ is the heat release time profile. Usually, $f(t)$ coincides with the time profile of the laser pulse, but since the pulse duration (380 fs) in our experiment is several orders of magnitude shorter than the electron-phonon relaxation time (hundreds of picosecond) observed by pump-probe technique (See details in Supplementary Note 8), here we adopted $f(t) = \Delta R(t)/R$.

A system of Eq. (5) has been solved numerically for the air surrounding ($z \leq 0$, $\chi_{air} = 18.70$ nm$^2$/ps, $q = 0$), VSe$_2$ ($0 < z \leq L$, $\chi_{VSe_2} = 1.392$ nm$^2$/ps, $q$ by Eq.(6)), and sapphire ($z > L$, $\chi_{sap} = 8.385$ nm$^2$/ps, $q = 0$), with the initial conditions $T(z,0) = 293$ K and the fourth-kind boundary conditions:[45] $T(0_-,t) = T(0_+,t)$, $T(L_-,t) = T(L_+,t)$, $T(-\infty,t) = T(+\infty,t) = 293$ K, $\kappa_{air}\partial T(0_-,t)/\partial z = \kappa_{VSe_2}\partial T(0_+,t)/\partial z$, $\kappa_{VSe_2}\partial T(L_-,t)/\partial z = \kappa_{sap}\partial T(L_+,t)/\partial z$. For a

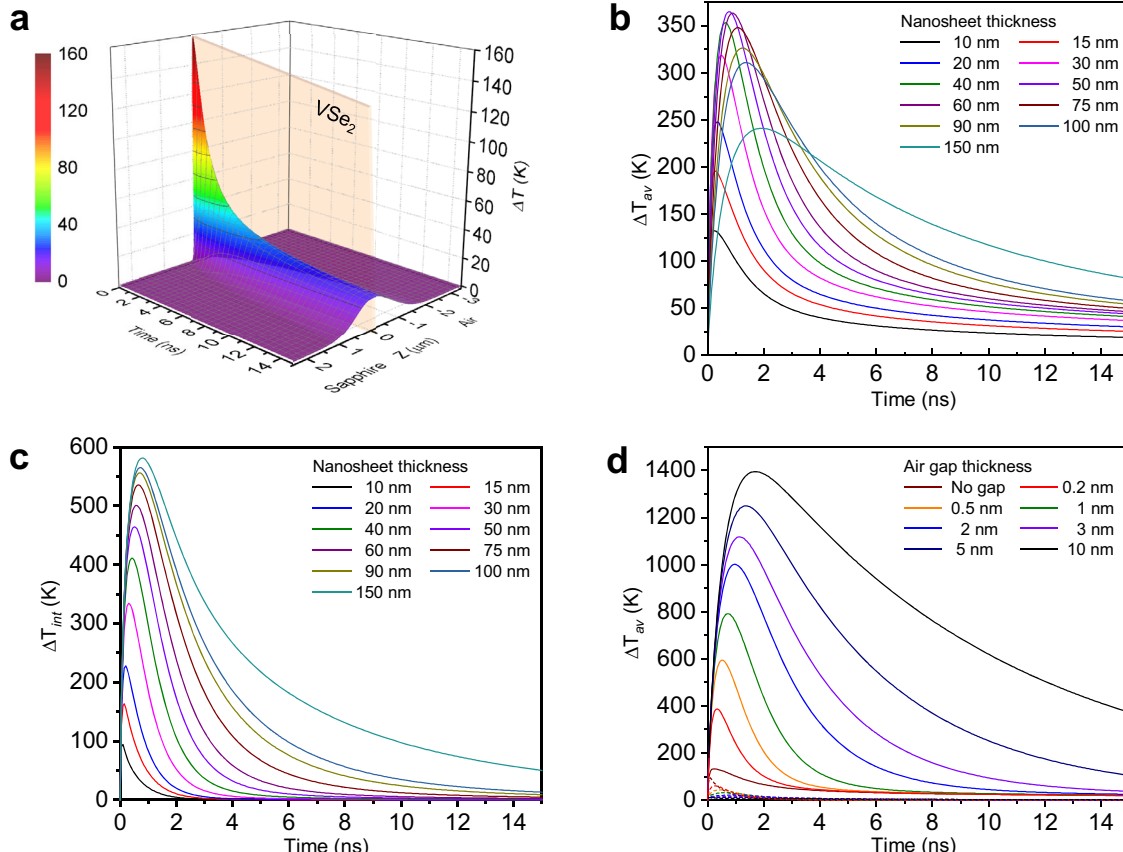

**Fig. 4 | Modelling of temperature increment space-time distributions through the VSe₂ nanosheet after a laser pulse. a** Map of general temperature increment above the room temperature ($\Delta T$) for $L = 10$ nm. **b** Average temperature increment of the VSe₂ layer ($\Delta T_{av}$) of different thicknesses. **c** Temperature difference between two interfaces of the VSe₂ layer ($\Delta T_{int}$) of different thicknesses. **d** Average (solid)

and interface temperature difference (dashed) of the 10 nm-thick layer ($\Delta T_{av}$) with an air gap between VSe₂ and sapphire of different thicknesses. The dashed lines are the corresponding temperature difference between top and bottom surfaces of the nanosheet.

VSe₂ nanosheet with $L = 10$ nm after the irradiation of a laser pulse with $E_p = 12$ nJ, the resulting distribution of the temperature increment above the room temperature $\Delta T(z,t) = T(z,t) - 293$ is shown in Fig. 4a.

Figure 4b shows the average temperature increment for nanosheets of various thickness: $\Delta T_{av}(t) = L^{-1}\int_0^L \Delta T(z,t)dz$. As the thickness increases, $\Delta T_{av}$ grows until $L \approx 50$ nm, then keeps a moderate value of around 350 K, and then decreases. We presume that the electron-phonon relaxation prolonged with increasing thickness (See details in Supplementary Note 8). This can reduce the thickness dependency, confirming again the fact that the thickness dependence of the nanosheet heating is quite modest.

The temperature difference generates a thermal stress in the nanosheet:[42]

$$\sigma_{th} = -B\beta_{TE}\Delta T \qquad (7)$$

where $B$ is bulk modulus and $\beta_{TE}$ is the volume thermal expansion coefficient with the average value: $\beta_{TE} = 2\alpha_{TE}^{\parallel} + \alpha_{TE}^{\perp}$, where $\alpha_{TE}^{\parallel}$ and $\alpha_{TE}^{\perp}$ are the in-plane and out-of-plane values of the linear thermal expansion coefficient, respectively. With the average values of these characteristics in Supplementary Table 1, it holds $\sigma_{th} = 0.127$ GPa per every 100 K of $\Delta T$. If we consider a non-uniform nanosheet with different thicknesses at different areas, we can still refer to Fig. 4b neglecting the in-plane thermal diffusion, because its relaxation time $\tau_{TR}^{\parallel} \sim a^2/\chi^{\parallel} \approx 4$ μs is incomparably larger than the corresponding out-of-plane relaxation time $\tau_{TR}^{\perp} \sim L^2/\chi^{\perp} \approx 72$ ps. Therefore, there will be a horizontal thermal stress in the non-uniform nanosheet, but its value

will be limited by 0.45 GPa, which is only slightly exceeds the friction shear stress. For nanosheets with $L = 10$ nm, the stress is quite insignificant.

The difference in the temperature of interfaces ($\Delta T_{int} = T(0,t) - T(L,t)$) is more remarkable and grows monotonously with increasing thickness, and its relaxation is noticeably faster than that of the average temperature, as shown in Fig. 4c. This can give a vertical thermal stress pulse; however, its value looks insufficient to overcome the vdW adhesion forces. Moreover, it cannot provide a horizontal movement of the nanosheet.

The situation changes dramatically when a narrow air gap appears at the interface between the nanosheet and the substrate. In this case, the calculation in Fig. 4d shows that the average temperature can jump up to ~1500 K depending on the thickness of the gap. The temperature difference between top and bottom surfaces of the nanosheet is insignificant (dashed curves in Fig. 4d). As it is demonstrated by the cross-sectional TEM images in Fig. 3 and Supplementary Note 4, such a situation occurs at the edges of a ME nanosheet, where these gaps are apparently formed during exfoliation due to the separation of the nanosheet from the bulk material with uneven delamination of the layers along the edges.

The height profiles obtained from AFM images of VSe₂ and TiSe₂ (See details in Supplementary Note 9) show that the thicknesses of the nanosheets are very heterogeneous and often become larger at the edges of the nanosheets, which is in agreement with the edge gaps: in the gap area the edges may be raised due to material fractures. The profiles of the CVD nanosheets are more even, however, there are also

protrusions at the edges, which indicate possible cracks in the contact. Therefore, local overheating of the edges is possible. In this situation, a thermal stress is generated (calculated by Eq. (7)), which is one order of magnitude larger than the friction shear stress and thus can effectively drive the nanosheet.

The thermal stress produces elastic waves in the nanosheet more intensive along the surface (Lamb waves). A strict consideration should include a complex boundary problem with reflection of the waves at the interface with sapphire where the inhomogeneous contact should be carried out. But to demonstrate the principal, we can consider a one-dimensional stress ($\partial\sigma/\partial y = \partial\sigma/\partial z = 0$), taking advantage of the facts that the direction in the plane of the nanosheet can be chosen arbitrarily, and the out-of-plane thermal difference is very small (Fig. 4d, dashed). Then in a free-standing nanosheet, longitudinal compression waves are induced, in which the $x$-component of the displacement $u$ is described by the one-dimensional wave equation:[42,46]

$$\frac{\partial^2 u(x,t)}{\partial t^2} = c_l{}^2 \frac{\partial^2 u(x,t)}{\partial x^2} + \frac{1}{\rho}\frac{\partial\sigma(x,t)}{\partial x} \qquad (8)$$

where the square of the longitudinal wave velocity is $c_l{}^2 = E/[\rho(1-\nu^2)]$[46], and the stress acting on the plate consists of thermal stress and the friction shear stress counteracting it: $\sigma(x,t) = \sigma_{th}(x,t) - \sigma_{fr}(x,t)$[23]. Since the former is much greater than the upper bound estimate of the latter, we restrict ourself to the case of motion without friction. Then Eq. (8) is conveniently solved by the d'Alembert formula[47], which under zero initial conditions and considering Eq. (7) will be written as:

$$u(x,t) = \frac{B\beta_{TE}}{2c_l\rho}\int_0^t \{\Delta T[x - \upsilon_l(t-\tau),\tau] - \Delta T[x + \upsilon_l(t-\tau),\tau]\}d\tau \qquad (9)$$

The free boundary conditions, $\partial u(0,t)/\partial x = \partial u(a,t)/\partial x = 0$, are generated by an odd expansion of the thermal stress function to the infinite axis according to the method of reflections: $\Delta T(2ka+x,t) = \Delta T(x,t)$, $\Delta T(2ka-x,t) = -\Delta T(x,t)$, where $k$ is an integer.

Here we modelled a partially bad contact by introducing an air gap of 10 nm thickness at one of the sides of the nanosheet of the length $a = 3.7\,\mu m$ (Fig. 5a). In our view, this corresponds to the situation of (I) an edge fracture, which happens in ME VSe$_2$ according to our TEM observation, and/or (II) a speck of the VSe$_2$ nanodust on the surface of the substrate that have broken off from the main sheet, which is also possible in some cases during coating. However, to keep the problem one-dimensional, we presumed that the gap is uniform in $y$ direction, like a pleat. The gap gradually vanishes to the center of the nanosheet's length, the half-thickness length of the gap is 1.5 $\mu$m. The corresponding thermal stress is mapped in Fig. 5b according to Eq. (7). Within the framework of the considered model, such stress produces two opposite longitudinal compression waves in the nanosheet shown in Fig. 5c. The particle displacement becomes asymmetric in the time scale, where positive $u$ values prevail over negative. The acoustic waves carry a momentum, which is generally defined as:[45]

$$\mathbf{M} = \frac{1}{c_l{}^2}\int \mathbf{q}\,dV = \frac{m_{NS}}{c_l a}\int_0^a \upsilon^2(x,t)\,sign[\upsilon(x,t)]dx \qquad (10)$$

where $\mathbf{q} = c_l\rho\upsilon^2\mathbf{n}$ is the volumetric density of the energy flux, $\upsilon = du/dx$ is the vibrational velocity of the travelling wave, $\mathbf{n}$ is the unity vector of the wave propagation direction, and $m_{NS}$ is the mass of the nanosheet. The ratio $M/m_{NS}$ is traced in Fig. 5d up to the time $t_{end} = 27.2$ ns, when the thermal stress becomes equal to the friction shear stress, and thus the motion should be stopped. During the considered period of time, the sum of two waves, in contact with the surface, will transmit a part of its momentum to the surface, which translates the nanosheet mass in the direction opposite to the direction of the wave momentum

(Fig. 6a). In the ideal case, the path of the nanosheet will hold: $\Delta x = \int_0^{t_{end}} M(t)dt/m_{NS} = 0.319$ nm. This is one third of our experimental value of 1 nm, obtained for such a nanosheet related to one pulse with $E_p = 12$ nJ (Fig. 2c).

It should be noted that a symmetric air gap at the two opposite edges of the same total length of 1.5 $\mu$m (Fig. 5e) induces a symmetric thermal stress (Fig. 5f), which results in a symmetric displacement (Fig. 5g), almost zero average momentum (Fig. 5h), and the path $\Delta x = 6.6$ fm (representing the level of calculation uncertainty). In this case, the speculative interaction with the substrate will be symmetrical in both directions, and the resulting thrust force will not appear (Fig. 6b). Thus, the asymmetry of poor contact is a determining factor in the photoacoustic mechanism of the nanosheet movement.

Our model ignores the antisymmetric Lamb waves as well as the real reflection conditions of the VSe$_2$/sapphire contact. We believe that further research on the nature of the nanosheet-substrate interaction and the corresponding 3D problem will enrich the consideration and describe this phenomenon more accurately. However, it already helps to understand some of the observed peculiarities at a semi-quantitative level. Thus, the efficiency of movement on different substrates can be considered from the point-of-view of their different thermal conductivities (See details in Supplementary Note 10). The analysis shows that in the case of the quartz glass substrate which possess a lower thermal diffusivity, the temperature increment is remarkably higher in the parts of the nanosheet which closely contact the substrate, but not in the free-standing edges. This leads to a lower asymmetry of the in-plane thermal stress and of the corresponding thermoelastic wave momentum. On the other hand, the friction force is weaker, that leads to large $t_{end}$, which depends on $\sigma_{fr}$ nonlinearly. According to our estimates, the possible value for the VSe$_2$-Quartz contact is $t_{end} \approx 255$ ns. The momentum oscillation asymmetry at $t \gg 1$ ns provides the constant speed of $\upsilon_{NS} = 2.8$ m/s, which results in the nanosheet's displacement of $\Delta x \approx 0.7$ nm. This explains a better efficiency of movement on quartz glass substrate than on sapphire substrate. The immobility on silicon substrate is harder due to the lower value of thermal diffusivity, but rather relates to the increased friction: $t_{end}$ shortens nonlinearly with even small increases in $\sigma_{fr}$, and $\Delta x$ becomes insignificant.

The thermoelastic effect described here also explains the importance of the pulse radiation regime for the optical driving. For the case of a CW radiation or a high-repetition radiation with the heat accumulation regime, the heating is slow compared to the establishment of thermal equilibrium along the nanosheet. Indeed, for the characteristic time of the latter, $\tau_{TR}^{\parallel}$, the maximal possible heating of the free-standing edges is $I\tau_{TR}^{\parallel}(1-R)(1-e^{-\alpha L})/\rho C_p L = 6.5$ K, which is not enough to produce the optical driving effect. In the subsequent time, thermal equilibrium is established in the nanosheet, and no asymmetry in the thermal stress and the related elastic wave momentum is appeared.

Nonuniform heating of the surrounding air could be another candidate for the drive by the photophoretic force. Photophoretic force can transport particles in ambient gases due to light-induced nonuniform heating[8,9]. However, upon a simple evaluation[9], the surface density of photophoretic forces: $F_{PP}/S \approx -4.3\times10^{-6}dT/dx = 6.4$ kPa (at a temperature gradient $dT/dx = -1.5\times10^9$ K/m corresponding to Fig. 5a), indicates insignificance of mechanisms of this kind in our system.

This conclusion is confirmed by other observations. In particular, when we decrease the repetition frequency of the fs laser to 1 Hz, the nanosheet still moves at a relatively slow speed, as shown in Fig. 7 and Supplementary Movie 3. Therefore, it moves with only a single ultrafast laser pulse acting on the nanosheet per second, where the time interval is sufficient for thermal relaxation of the nanosheet and the environment. Additionally, as we mentioned above, CW laser, which is profitable for the convection processes, cannot achieve the optical drive.

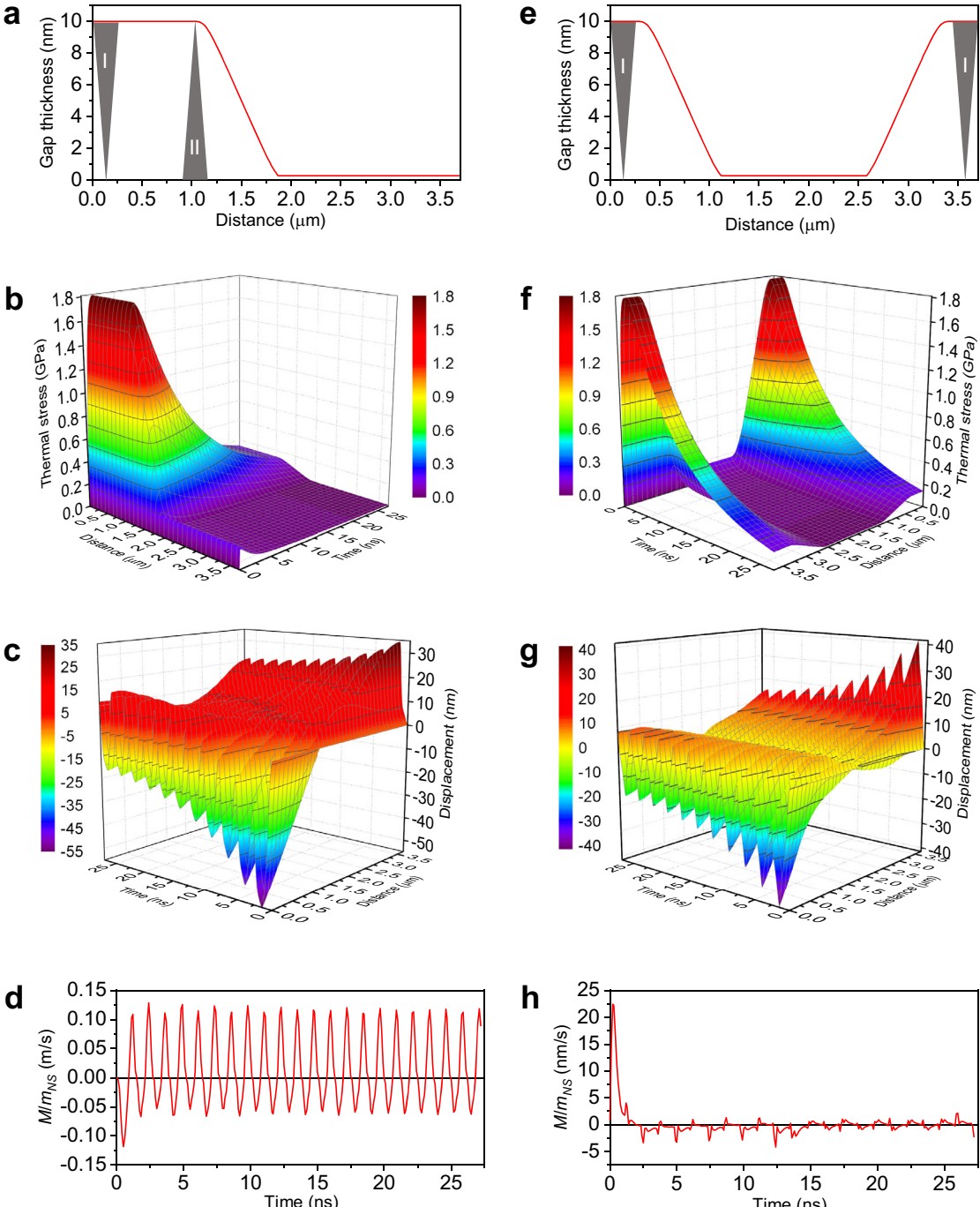

**Fig. 5 | Modelling of thermoelastic effect in a free-standing VSe₂ nanosheet with *L* = 10 nm and *a* = 3.7 μm following a laser pulse with *Eₚ* = 12 nJ. a** Illustrations of asymmetric air gap profile to the substrate. I: the case of an edge fracture, II: the case of a speck of dust on the surface of the substrate. **b** Thermal stress map in the case of asymmetric air gap. **c** Maps of longitudinal compression wave in the nanosheet in the case of asymmetric air gap. **d** Dynamic momentum-to-mass ratio

($M/m_{NS}$) in the case of asymmetric air gap. **e** Illustrations of symmetric air gap profile to the substrate. **f** Thermal stress map in the case of symmetric air gap. **g** Maps of longitudinal compression wave in the nanosheet in the case of symmetric air gap. **h** Dynamic momentum-to-mass ratio ($M/m_{NS}$) in the case of symmetric air gap.

We also note in Fig. 7 that the nanosheet can pass one on top of another, which confirms our view of the thermoelastic mechanism, because such movement inevitably increases the air gap with the substrate and therefore facilitates the motion.

**All-optical splicing of 2D nanosheets**

The achieved all-optical driving of 2D nanosheets in non-fluidic environment can inspire its practical applications[48]. For example, as

shown in Fig. 8 and Supplementary Movie 4, a separated nanosheet gradually moves towards the larger one until they were successfully pieced together. Consequently, the photoacoustic manipulation method allows us to assemble 2D materials together and build out structures of them, which is a promising way to fabricate the lateral photonic and optoelectronic devices. In particular, semimetal 1T-VSe₂ and 1T-TiSe₂ are promising materials for 2D metal electrodes applied in 2D optoelectronic devices, such as field-effect transistors and

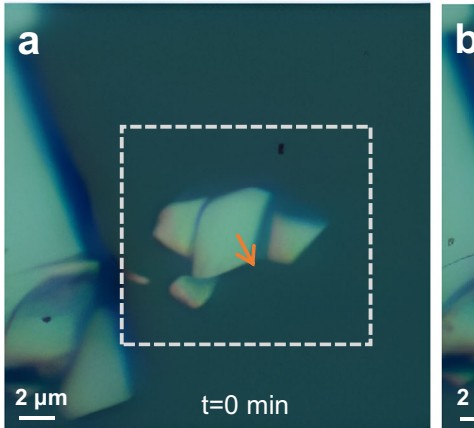

**Fig. 6 | Schematic of the generated photoacoustic forces in a nanosheet on the substrate. a** The case of asymmetric stress: the acoustic waves carry a resulting nonzero momentum (**M**) which is transferred to the substrate producing the unidirectional traction force ($F_{traction}$) that drives the nanosheet to move with a velocity ($v_{NS}$). **b** The case of symmetric stress: the resulting moment of two opposite acoustic wave and the traction forces are compensated.

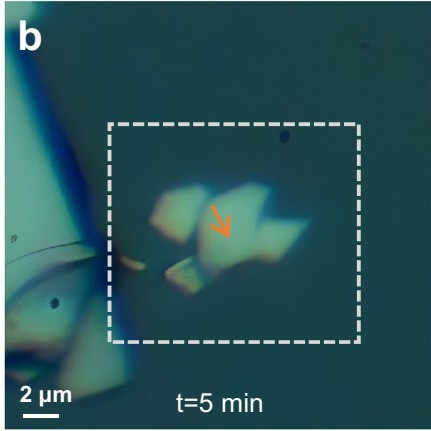

**Fig. 7 | Optical drive for VSe₂ nanosheets at the frequency of 1 Hz. a, b** The state of VSe₂ nanosheets at 0 min and 5 min. The arrows represent the moving direction from 0 to 5 min. The laser source is the fs pulsed laser at 1040 nm with the repetition frequency of 1 Hz (See Supplementary Movie 3).

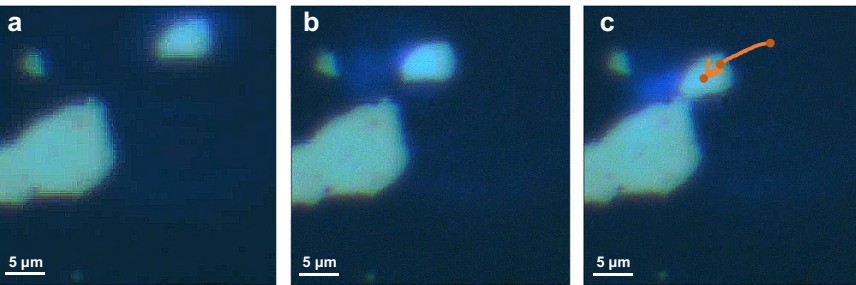

**Fig. 8 | All-optical splicing of VSe₂ nanosheets. a–c** Optical micrographs of nanosheet assembly process under the control of femtosecond laser at 1040 nm with the repetition frequency of 1 kHz (See Supplementary Movie 4). The arrow draws the trace curve.

photodetectors[49–51]. Compared with traditional 2D layered semiconductors and 3D metal electrode configurations, the configuration of 2D semiconductors and 2D electrodes can significantly reduce the contact resistance at the interface, which can effectively improve the device performance[44,52]. Therefore, the photoacoustic actuating system provides some insight into the manufacturing technique of the photonic and optoelectronic integrated circuits.

In summary, we have found 2D nano-actuator systems capable of overcoming the large vdW interaction between sample and substrate. VSe₂ and TiSe₂ nanosheets prepared by ME method are readily moved on the horizontal sapphire substrate plane by simple downward laser irradiation. The nanosheets can be also moved on quartz substrate with a lower adhesive force, and cannot be moved on silicon substrate with a larger adhesive force. The maximal speed efficiency of nanosheet manipulation achieved in our experiment is 434 μm·s⁻¹ per milliwatt, which is even larger than previously demonstrated in other experiments with gold nanoplates on microfibers. The moved 2D

materials belong to semimetals and differ in large values of absorption and thermal expansion coefficients. 2D materials with essentially lower absorption coefficients such as BN, MoS₂, WSe₂, and PdSe₂ do not demonstrate sensitivity to the optical drive. We assign the phenomena to the thermoelastic mechanism of surface acoustic wave generation. Numerical study of the spatial and temporal distributions of the heating temperature attained upon laser irradiation reveals the possibility of the generation of an essential thermal stress in the nanosheet in the case of an air gap between the nanosheet and the substrate at the nanosheet edges. We demonstrate that the longitudinal surface waves resulting from this thermal stress carry a significant momentum that is sufficient to move the nanosheet along a horizontal surface at a speed close to that observed in the experiment. The mechanism is essentially pulsed, since it needs nonequilibrium (gradient) heating of the nanosheet for its operation. Other possible mechanisms, including optical pressure, the photoacoustic mechanisms except thermoelastic force, and photophoretic force, appear negligible in this optical drive.

Based on these observations, this actuator system has the potential for applications in touch-free and non-invasive transfer, such as the preparation of lateral heterostructures. It is also a good tool to study the vdW interactions between 2D materials and substrate. The development of the optical drive in vdW environments will continuously reveal more physical phenomena in light-matter interaction and motivate further studies in both theoretical models and device applications in optical, mechanics, nanotechnology, and biology.

## Methods

### Preparation and characterization

The 2D nanosheets were mechanical exfoliated from bulk single crystals and then transferred to the polished sapphire, IR quartz glass, and silicon substrates. Nanosheet height-profile measurements were performed using atomic force microscope FM-Nanoview 6800 operated in the tapping mode. Surface quality of the substrates was checked using Veeco Instruments MultiMode V atomic force microscope. Raman characterization was carried out using a confocal microscopy system (LabRAM HR Evolution) excited by 532 nm CW laser. Cross section of the $VSe_2$-sapphire contact was prepared using Helios 5 UX DualBeam combining a FIB and a scanning electron microscope. Before FIB cutting, the sample was coated with chromium via sputtering. High-resolution TEM images were carried out using Talos F200X instrument.

### Optical drive system

The laser source was set at 1040 nm with a pulse width of 380 fs. The repetition frequency, $f_{rep}$, can be tunable from 1 Hz to 100 kHz. The laser beam was reshaped via an apodizing filter from a Gaussian beam into a near-flat beam with constant intensity. Then the laser beam was introduced into a microscope and the sample was placed on the horizontal sample stage. Finally, the laser was perpendicularly focused on the surface of the $VSe_2$ nanosheets via an objective lens (NA = 0.5, 50×), and the spot radius was approximately 5 μm. The movements of $VSe_2$ nanosheets was captured by a CCD camera. The illustration of our optical drive system is shown in Supplementary Fig. 1.

### Pump-probe system

The ultrafast pump-probe measurements were performed using the same laser source (1040 nm, 380 fs, 10 kHz). The pump and probe pulses were split from the laser source by an ultrafast beam splitter. Either pump or probe beam was doubled to 520 nm, as required. The pump beam was modulated by a chopper, collinearly combined with the probe beam and delivered into the microscope. The laser beams were focused onto the sample with the same 50× objective lens. The reflected beams were passed through a filter to block the pump beam, and was then directed into a silicon detector to obtain differential-reflection signal $\Delta R/R$. The illustration of our pump-probe system is shown in Supplementary Fig. 7.

### Computational methods

Heat equation system has been solved by the finite difference method using an implicit four-point algorithm. The step along the spatial coordinate was chosen $dz = 0.1$ nm, and along the time $dt = 0.5$ ps. The error associated with the finite step quantity was less than 0.5%. The time decay of the averaged over nanosheet thickness temperature difference $\Delta T_{av}(t)$ was fitted by three exponents to extrapolate the temporal dependence of the thermal stress to a larger time range. The integral in Eq. (9) has been computed by the Simpson method with the step $d\tau = 1$ ps. The ($\Delta x$, $\Delta t$) mesh in Fig. 5c, g is (13.6 nm, 100 ps).

## Data availability

Relevant data supporting the key findings of this study are available within the article and the Supplementary Information file. All raw data generated during the current study are available from the corresponding authors upon request.

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

## Acknowledgements

This work is supported by National Natural Science Foundation of China (NSFC) (No. 61975221 and 12174414, to J.W.), the Strategic Priority Research Program of CAS (No. XDB43010303, to J.W.), and High-End Foreign Expert Fellowship of the Ministry of Science and Technology of the People's Republic of China (No. G2021061005L, to I.M.K.).

## Author contributions

J.W. conceived the original idea. J.W., I.M.K., and X.C. led the project and wrote the paper. X.C. designed and built the optical drive system and microscopic pump-probe system. X.C. performed the experiments, collected, and analyzed the data. I.M.K. performed the theoretical calculations and analysis. Y.X. and Y.W. contributed in material fabrication. T.W. and L.Z. helped in data analysis. All authors discussed the results and commented on the manuscript.

## Competing interests

The authors declare no competing interests.
