## [Peer Review File · Nature Communications]

Photoacoustic 2D actuator via femtosecond pulsed laser action on van der Waals interfacesREVIEWER COMMENTS

Reviewer #1 (Remarks to the Author):

The paper by Xi Chen et al. reports optical actuators of 2D nanosheets driven by ultrafast pulses on vdW interfaces. Such actuators show fine locomotion performance with speed efficiency characterized to be $66.5 \mu\text{m} \cdot \text{s}^{-1} \cdot \text{mW}^{-1}$, and their motion directions are controlled by movement directions of laser pulses. The results could be of interest to the micro-actuator community.

However, the concept of using laser-induced acoustic/elastic waves to enable locomotion of micro-objects on vdW interfaces is not new. It has already been proposed by a few recent publications, such as *Nat. Commun.* **12**, 385 (2021) and *Sci. Adv.* **5**, 8271 (2019), which have also been cited by the present paper. More specifically, comparing with these existing findings, the reported results show primary differences in two aspects: (1) using VSe₂ nanosheets instead of gold microplates; (2) placing nanosheets on sapphire substrates instead of on micro-fibers. Regarding this, the conceptual novelty of the paper is limited. On the theoretical side, the analysis of the underlying mechanisms lacks of rigor and even contains some mistakes [specifically in Eqs. (4) and (9), see detailed comments below]. On the experimental side, more details concerning the experimental results should be given at least in Supplementary Materials.

Overall, the paper seems interesting, and the referee, thus, believe that the paper can be published in some specific journals. However, the referee *cannot recommend publication in Nature Communications* due to the mentioned reasons above.

Coming to the details of the paper, the referee provides the following comments.

1. Eq. (4) is not used properly for evaluating force experienced by nanosheets. This can be directly examined by considering a specific case assuming that there is no real-contact between substrates and nanosheets; accordingly, the force should be zero. However, with Eq. (4), the force is nonzero considering $|\text{Rac}|=1$ for a non-contact interface. The correct one should include contributions from both incident and reflected acoustic waves.
2. Eqs. (8) and (9), which explain motion speeds of nanosheets, are not used properly. The translation of nanosheets essentially requires the movement of the laser. If the laser is still and the pulse impinges on the nanosheets uniformly (symmetrically), the nanosheets cannot move. Therefore, the correct formulations should include these factors, which, however, are absent in Eqs. (8) and (9). Moreover, Eq. (8) is adopted from Ref. [36], and the authors should carefully check the applicable conditions of this equation and its physical meaning.
3. The authors estimated the dry adhesion force between the VSe₂ and substrate to be 4.7 mN. I doubt that it may be wrong and overestimated. The authors use a contact distance of 0.2 nm. I have serious concerns about this contact distance used. I would say, firstly, even though the material and the substrate are the same, the

contact situation between the VSe₂ and the substrate can be totally different if the transferring method or situation alters. Secondly, even if the authors take the same cross-sectional transmission electron microscopy (TEM) image for their own samples to measure the contact distance, the results may be also not correct, since the FIB (or another method for the sample preparation) would change the original contact situation between the VSe₂ and substrate when preparing the sample for the TEM. Thirdly, according to the AFM characterization results in Fig. 1d, the roughness of the VSe₂ sheet and the substrate is in the nanometer scale. It seems the roughness has 2 nm by roughly judging from the insert curve profile in Fig. 1d. If using 2 nm as the contact gap, the estimated adhesion force will be 4.7 μm , which means that the authors may overestimated the adhesion force by 3 orders.

Reviewer #2 (Remarks to the Author):

The work by Chen et al. exploits the femtosecond laser to directly actuate and manipulate VSe₂ nanosheets in nonliquid environment. This is an interesting work that 2D materials can be directly manipulated by light although the experimental result is rather elementary. The authors propose that the underlying mechanism is mainly derived from photoacoustic effects: the generated longitudinal depthward acoustic waves (DAW) contributes to overcoming the substrate adhesion, while the propagation of transverse surface acoustic waves (SAW) accounts for the horizontal movement of the nanosheet. Nevertheless, there are still some technical points needed to be addressed:

1. The authors investigated the light-actuated moving speed of the nanosheet as functions of the laser repetition frequency and the pulse energy, given in Fig.3. It will be nice to understand the relationship between moving speed and the thickness of the nanosheet for better understanding the physical mechanism.
2. The authors mainly compared the manipulation behavior between VSe₂ and MoS₂ nanosheets. How about other 2D materials? More discussions are expected to explore the material dependency.
3. As the acoustic reflection coefficient of the nanosheet-substrate interface plays a key role in light-actuation. What will happen on different substrates?
4. Is the nanosheet suspended above the substrate or still weakly adhered to the substrate after light-actuation? If the nanosheet is adhered to the substrate, what's the magnitude of the friction force between the nanosheet and the substrate?

Reviewer #3 (Remarks to the Author):

This manuscript reports an optical manipulation method to transport single VSe₂ nanosheets on a polished sapphire substrate in air. Considering that most of the previous studies on optical manipulation of micro-/nanostructures were done in liquid and few optical methods could overcome the Van Der Waals interactions, this work shows a good level of novelty and provides a new method to the optical manipulation community based on the photoacoustic effects. The major weakness of this work, however, is that this method seems to be limited to a specific system (i.e., VSe₂ nanosheets on a sapphire substrate) and cannot be generalized to other nanostructures (e.g., MoS₂ nanosheets could not be moved in this work). If the authors can demonstrate or provide ways to generalize this method, this work will certainly be more attractive.

Another weakness is that the manuscript attributed the mechanism to the photoacoustic force after excluding other possible mechanisms. However, it lacks direct experimental evidence of the existence of the photoacoustic force and characterization of its features. The discussions were based on literature, but the conditions in the literature were not necessarily the same. The authors can consider some pump-probe experiments to make the photoacoustic mechanism more convincing. Note that the acoustic vibrations of nanostructures on substrates have been studied by ultrafast pump-probe spectroscopy, e.g.,

1. Yu, K.; et al., Damping of Acoustic Vibrations of Immobilized Single Gold Nanorods in Different Environments. *Nano Letters* 2013, 13 (6), 2710-2716.
2. Wang, J.; et al., Long Lifetime and Coupling of Acoustic Vibrations of Gold Nanoplates on Unsupported Thin Films. *The journal of physical chemistry. A* 2019, 123 (47), 10339-10346.

Some other comments are listed below.

1. The abstract stated that "The speed efficiency ... exceeds that of optical trapping by three orders of magnitude." This is not accurate. The conclusion is based on results in Ref. 50, but that work is not the normal "optical trapping" but rather a special experiment that used propagating surface plasmons to transport nanoparticles. Traditional optical traps can already move nanostructures with high speeds, and the modern phase gradient optical traps can drive nanostructures even faster.

2. The manuscript used a contact distance $d=0.20\text{nm} \pm 0.05\text{nm}$ for calculations based on the results in Ref. [26]. That paper studied VSe₂ thin films synthesized by molecular beam epitaxy on sapphire substrates. The contact distance of exfoliated nanosheets on a polished sapphire

substrate in this work is likely much larger than that, especially considering the roughness of the substrate and the result shown in Fig. 1d.

3. The manuscript said, "the Raman spectrum from 100 to 300 cm^{-1} under 532 nm excitation is presented in Fig. 2c" in the DISCUSSIONS but "Raman characterization was ... excited by 633 nm cw laser." in METHODS. Which wavelength is correct?

4. In Fig. 2, the laser spot of 5 μm radius should cover all the four nanosheets in the blue square, why was only the middle one moved?

5. How did the authors measure the results in Fig. 3? Did they move the laser spot? The maximum distance was larger than 10 μm so it is likely that the laser spot was moved. How would that affect the measurement? In addition, if the laser spot was moved, optical trapping force will play a role in the movement of the nanosheet because a large intensity gradient exists at the edge of the pattern even for a flat-top beam.

6. A major question is how the laser controls the moving direction of the nanosheet? If the laser fully covers a nanosheet, why is the acoustic wave directional? The nanosheet in Fig. 5 apparently moved in a curve, how would that happen?

7. How was the "escape momentum of the vdW force" calculated?

8. The manuscript said, "the initial offset energy derived from the horizontal intercept in Fig. 3b is approximately 7.8 nJ". Why didn't the authors experimentally demonstrate that by tuning the laser pulse energy to identify the threshold?

REVIEWER COMMENTS

Reviewer #2 (Remarks to the Author):

In the revised manuscript, Chen et al. provided more detailed information on this experiment and addressed part of my concerns. It is nice to see that the technique can be applied to different materials. However, I still have some concerns on this work:

1. In pages 5-6, the authors claimed that continuous-wave laser cannot drive the motion of the nanosheet but did not explain in detail. In my opinion, CW laser may also induce the overheating in the edge of nanosheet to drive its motion if the mechanism is correct. I expect more discussions here.
2. In page 6, what's the pulse width of the "quasi-continuous laser"? It is inappropriate to call the laser with high repetition rate as quasi-continuous laser if its pulse width is short. The authors need to double check this term.
3. The authors claimed that they cannot characterize the separation distance between the substrate and the nanosheet but provided a calculated value in the case of ideal contact. Determining the separation distance in experiments is important for the understanding of underlying mechanism. The authors may characterize it by subtracting the calculated thickness of 2D material from the height of tested sample.
4. In Table R2, the thermal conductivity of quartz glass is much lower than that of sapphire. Thus, the temperature of the nanosheet or the interfacial temperature difference may be significantly different on various substrates, which may influence the motion of the nanosheet. The authors may want to comment on this.

Reviewer #3 (Remarks to the Author):

In the revised manuscript, the authors added experimental results and theoretical analyses, some were based on my suggestions, which are appreciated. I believe the experimental results are interesting and the theoretical analyses are valuable, however, the connections between the two are weak. That is, the theoretical analyses are largely based on assumptions and indirect evidence inferred from the experimental results. They can more or less serve the role to explain the experimental observation, but the assumptions seem to lack solid basis and contain too many estimations. This is the major weakness of the manuscript.

Some more comments:

1. I agree with the Reviewer 1 on the concern of the calculation of vdW force, especially the 0.2 nm contact distance. On the one side, in Supplementary Note 3, the authors measured the roughness of the substrates by AFM. Those substrates should be cleaned substrates, but after transferring the nanosheets, the substrates apparently became dirty (or contaminated) as shown in Supplementary Note 7, and the roughness would be much larger. On the other side. The authors used an atomic model to get a 0.255 nm distance, that would be an ideal case and may exist only for some contact points, in that case, how do the authors choose S in the equation 1?
2. About the optical force, note that the standard equation for gradient force is for a CW laser. For a fs pulsed laser, the equation would be different, and the authors may also need to consider nonlinear optical interactions. In addition, the authors used a dipole model but I don't think it is appropriate to approximate the microscale sheets as dipoles. The authors said "it has been previously shown that the gradient optical force is less sensitive to the particle size, so that even at a $\sim \lambda$ it gives a result that differs from that of the more rigorous Generalized Lorenz-Mie theory by no more than one order of magnitude.³⁵" This doesn't appear to be not a solid argument to me.

Responses to Reviewers

Review #1:

The paper by Xin Chen et al. reports optical actuators of 2D nanosheets driven by ultrafast pulses on vdW interfaces. Such actuators show fine locomotion performance with speed efficiency characterized to be $66.5 \mu\text{m} \cdot \text{s}^{-1} \cdot \text{mW}^{-1}$ and their motion directions are controlled by movement directions of laser pulses. The results could be of interest to the micro-actuator community.

However the concept of using laser-induced acoustic/elastic waves to enable locomotion of micro-objects on vdW interfaces is not new. It has already been proposed by a few recent publications such as *Nat. Commun.* **12**, 385 (2021) and *Sci. Adv.* **5**, 8271 (2019), which have also been cited by the present paper. More specifically, comparing with these existing findings, the reported results show primary differences in two aspects: (1) using VSe₂ nanosheets instead of gold microplates; (2) placing nanosheets on sapphire substrates instead of on micro-fibers. Regarding this, the conceptual novelty of the paper is limited. On the theoretical side, the analysis of the underlying mechanisms lacks of rigor and even contains some mistakes [specifically in Eqs. (4) and (9), see detailed comments below]. On the experimental side, more details concerning the experimental results should be given at least in Supplementary Materials.

Reply: We would like to thank the reviewer for the appreciation of our work expressed here and for the subsequent valuable comments. Your comments are very insightful and are very helpful to improve our manuscript.

We studied more 2D materials in the latest experiments, including h-BN, MoS₂, WSe₂, PdSe₂, and TiSe₂. We found TiSe₂ performed similar motion behavior with VSe₂. Therefore, we believe that this is not the exceptional behavior of VSe₂ nanosheets, but the property inherent in a certain class of materials. Especially, VSe₂ and TiSe₂ belong to semimetals, and they have quite large absorption coefficients and thermal expansion coefficients. Thus, we think that these two parameters are important characteristics of materials that can move under laser drive.

Concerning the conceptual novelty, we see more differences from the cited publications. This is the geometry of excitation: when we irradiate the nanosheet uniformly and perpendicular to its surface, we exclude some propelling mechanisms, such as surface plasmon and gradient fields in large part. Then the use of a femtosecond laser instead of continuous-wave or nanosecond one.

This is important for observing the phenomenon because it is not observed with continuous irradiation. These differences indicate the different nature of the excitation of Lamb waves in our case. Our report shows that even under these circumstances, certain materials can be set in motion by a femtosecond laser beam.

Concerning the theoretical calculations, we tried to correct the mistakes made and to enrich our theoretical consideration. Firstly, we reviewed our estimates of van der Waals forces and added an upper bound calculation for the friction force. Secondly, we added the temperature field calculations and the kinetics of the thermal stress in the system to provide a more detailed and accurate description of our experimental phenomenon. We have added these theoretical calculations in the revised manuscript.

Concerning the experimental part, we added more experimental details in the revised manuscript and provided a schematic diagram of our optical drive system in Supplementary Information. Furthermore, we added more experiments to enrich our work. Firstly, we compared different 2D materials (h-BN, MoS₂, WSe₂, PdSe₂, TiSe₂, and VSe₂) and different substrates (sapphire, quartz, and silicon), which can help us to raise a more systematic discussion about the optical manipulation of 2D materials. Secondly, we added more AFM profiles of the nanosheets and carried out a roughness analysis of the substrates performed with a higher resolution, which can provide evidence to our calculations of friction force and thermal stress. Thirdly, ultrafast carrier dynamics of VSe₂ nanosheets were carried out to provide information of its electron-phonon scattering and relaxation time. We have added these new experimental results and related discussions in the revised manuscript and Supplementary Information.

1. Eq. (4) is not used properly for evaluating force experienced by nanosheets. This can be directly examined by considering a specific case assuming that there is no real-contact between substrates and nanosheets; accordingly, the force should be zero. However, with Eq. (4), the force is nonzero considering $|\text{Rac}|=1$ for a noncontact interface. The correct one should include contributions from both incident and reflected acoustic waves.

Reply: We thank the reviewer for pointing put this issue.

Indeed, the accurate expression for the zero-angle sound pressure-amplitude reflection is $R = (Z_2 - Z_1)/(Z_2 + Z_1)$ [e.g. Eq.(3.3.4) in Ref. R1]. And we were wrong in that by introducing the sound reflection coefficient into Eq. (5) we took only the pressure of the reflected sound wave P_{refl} into account, whereas the momentum of the nanoparticle is given by a change in sound pressure upon reflection:

$$\Delta P = P_{\text{inc}} + P_{\text{refl}} = (1+R)P_{\text{inc}} = 2Z_2P_{\text{inc}}/(Z_2+Z_1).$$

Since $P_{\text{inc}} = \Delta P_{DAW}$ in our manuscript, the Eq. (5) should be:

$$F_{DAW} = 2Z_2\Delta P_{DAW}S/(Z_2+Z_1).$$

It triples the resulting force for VSe₂ on sapphire and at the same time makes the reflection coefficient a less significant factor in evaluations of the photoacoustic force. However, in the revised manuscript, realizing the complexity of the problem including its strong dependence on the nanosheet thickness and quality of the contact with substrate (see comments below), we do not consider the depthward acoustic waves and their effect. We leave this consideration for future.

2. Eqs. (8) and (9), which explain motion speeds of nanosheets, are not used properly.

The translation of nanosheets essentially requires the movement of the laser. If the laser is still and the pulse impinges on the nanosheets uniformly (symmetrically), the nanosheets cannot move. Therefore, the correct formulations should include these factors, which, however, are absent in Eqs. (8) and (9). Moreover, Eq. (8) is adopted from Ref. [36], and the authors should carefully check the applicable conditions of this equation and its physical meaning.

Reply: we thank the reviewer for this valuable comment.

We can see the nanosheet moving both when the laser is moving and when the laser is stationary. The former is easy to understand because the movement of laser create an unbalanced force to actuate the nanosheet, but the latter requires a deeper understanding. The theory given in Ref. [36] (now Ref. [43]), considers the generation of surface waves by a stationary laser beam. A special solution can be found when the beam is moving with the speed comparable with the Rayleigh wave speed, but this is far from our case. However, Eq. (8) refers to the vibrational velocity of the

medium particles in the Rayleigh wave and does not quite fit our case indeed, because, as we now understand, Lamb waves, not Rayleigh waves, propagate through the nanosheet, and other boundary conditions must be considered for it. This problem looks quite difficult to solve quickly, so in our revised manuscript we limit ourselves to considering a simple model of a longitudinal wave in a suspended thin film to understand the order of possible magnitude of the effect.

Your comment about the uniformity of irradiation and the absence of a certain direction of the movement is very important. Initially, we believed that the direction of motion of the generated surface wave is determined by random factors, such as fluctuations in optical absorption and adhesion forces, as well as the shape of the nanosheet. Now we understand that these are clearly insufficient reasons. Therefore, by numerical solving a one-dimensional heat equation, we obtained the temperature fields in irradiated VSe₂ nanosheets, which outlined direction and magnitude of the temperature differences in them. For example, Fig. R1a shows the result for the average excess temperature ($\Delta T_{av} = \langle T(t) \rangle_L - T_{room}$) of nanosheets of 10 nm and 100 nm thicknesses (solid curves), and the temperature difference between their interfaces (dashed curves, $\Delta T_{int} = T_{Air-VSe_2} - T_{VSe_2-Sap}$) after a 12-nJ-laser pulse.

Fig. R1 Calculated temperature differences followed after absorption of a 12-nJ-laser pulse. **a** Temperature differences in VSe₂ nanosheets with 10 nm and 100 nm thicknesses on a sapphire substrate; **b** Temperature differences in a 10-nm-thick nanosheet with a 5-nm-thick air gap between the nanosheet and the substrate.

As we can see from Fig. R1a, the heating of the nanosheet is small, especially for thicknesses of the order of 10 nm. But even with an increase in the thickness of the nanosheet (albeit local), the temperature difference in the plane of the nanosheet (horizontal direction) will not be too large. The corresponding thermoelastic stress is:^{R2}

$$\sigma_{TE} = -B\beta_{TE}\Delta T,$$

with $B\beta_{TE} = 1.25$ MPa/K (Table R2) will not exceed 0.25 GPa. At the same time, our estimates of friction forces exceed this value (see comments below). Therefore, we conclude that random thickness variation in a nanosheet cannot be the reason of its movement. Also, vertical stress will not be able to overcome van der Waals forces, at least for nanosheets of the order of 10 nm thick.

However, if we consider a gap between the nanosheet and the substrate, we see that sufficiently large differences occur (Fig. R1b). We can imagine such gaps in some places of the sample (mainly along the edges) because our AFM observations indicate corresponding profile curvatures in many cases (see Supplementary Note 7 in Supplementary Information file). According to our ideas, such gaps may appear due to either random foreign objects on the substrate (dust), or as a result of fractures of the edges of the nanosheet itself. Schematically, these cases are shown in Fig. R2.

Fig. R2 Schematic representation of the gap between the nanosheet and the substrate. **a** The gap is due to a dust nanoparticle; **b** The gap is due to a nanosheet fracture.

Currently, we are inclined to believe that this is the most likely reason that leads to the appearance of thermoelastic surface waves in nanosheets and their movement on the surface of the substrate. In the revised manuscript, we give the scheme of the occurrence of a thermoelastic wave driving nanoparticle and some discussions in “3.1 Nanosheet thermal analysis” and “3.2 Thermoelastic effect” subsections.

3. The authors estimated the dry adhesion force between the VSe₂ and substrate to be 4.7 mN. I doubt that it may be wrong and overestimated. The authors use a contact distance of 0.2 nm. I have serious concerns about this contact distance used. I would say, firstly, even though the material and the substrate are the same, the contact situation between the VSe₂ and the substrate can be totally different if the transferring method or situation alters. Secondly, even if the authors take the same cross-sectional transmission electron microscopy (TEM) image for their own samples to measure the contact distance, the results may be also not correct, since the FIB (or another method for the sample preparation) would change the original contact situation between the VSe₂ and substrate when preparing the sample for the TEM. Thirdly, according to the AFM characterization results in Fig. 1d, the roughness of the VSe₂ sheet and the substrate is in the nanometer scale. It seems the roughness has 2 nm by roughly judging from the insert curve profile in Fig. 1d. If using 2 nm as the contact gap, the estimated adhesion force will be 4.7 μm, which means that the authors may overestimated the adhesion force by 3 orders.

Reply: we thank the reviewer for this very insightful comment.

(1) A large error is possible indeed on the adhesive force due to the third power of the contact distance, but the error is still not by three orders of magnitude. The fluctuation of nanometer scale observed in the profile in Fig. 1d originates from our instrumental noise, rather than the roughness of the substrate. We examined the surface quality of the sapphire substrate with a more sensitive instrument (Veeco Instruments MultiMode V), as shown in Fig. R3. It can be found that its roughness is indeed within 0.2 nm, which is consistent the quality specified in the product certificate by its manufacturer ($R_a < 0.2$ nm).

In the revised manuscript, we added the roughness analysis of substrates in Supplementary Note 3.

Fig. R3 The roughness analysis of sapphire substrate by AFM.

(2) On the other hand, we also introduced an estimate of the contact distance d based on the rigid sphere model. The distance d can be regarded as the distance between centers of the lowest spheres of the top surface and the highest spheres of the bottom, as shown in Fig. R4.

Fig. R4 Schematic arrangement of a selenium atom (orange ball) of the VSe₂ nanosheet on the sapphire surface formed by oxygen atoms (blue balls).

The hard sphere radii of the surface atoms (oxygen from the sapphire side and selenium from VSe₂ side) are taken as half minimal O – O and Se – Se distances in the contacting planes, respectively: $r_O = 0.126 \text{ nm}$ ^{R3} and $r_{Se} = a_{lat}/2 = 0.168 \text{ nm}$ ^{R4} (where a_{lat} is the crystal lattice parameter). Oxygen atoms in the contact surface of *c*-sapphire form a face of hexagonal close-packed lattice,^{R5} and with a tight package, atoms of the top surface should lie in hollows of equilateral triangles formed by oxygen atoms.^{R6} Then it is easy to find $d =$

$\sqrt{(r_0 + r_{se})^2 - 4r_0^2/3} = 0.255 \text{ nm}$. Surprisingly, it turned out to be close to the magnitude observed in the TEM image ($d \approx 0.2 \text{ nm}$). With this value of the contact distance, the adhesion strength is $F_{vdW} \approx -2.3 \text{ mN}$. It is only two times less than our initial estimate (4.7 mN). However, we understand that this assessment is suitable for the case of ideal contact, which apparently takes place when the nanosheet is deposited by molecular beam epitaxy. In the case of mechanical exfoliation method, this force will be reduced due to a less optimal relative arrangement of the lattices of the top and bottom surfaces. We cannot estimate this value, so we limit ourselves to the upper bound estimate in the revised manuscript. Even though it has some estimation errors, but it can be evident enough for understanding the order of magnitude of van der Waals forces and then the friction forces. In particular, when considering the thermoelastic effect, it allows us to neglect friction.

In the revised manuscript, we added the above calculations and arguments in “1. Van der Waals adhesion and friction forces” subsection.

Review #2

The work by Chen et al. exploits the femtosecond laser to directly actuate and manipulate VSe₂ nanosheets in nonliquid environment. This is an interesting work that 2D materials can be directly manipulated by light although the experimental result is rather elementary. The authors propose that the underlying mechanism is mainly derived from photoacoustic effects: the generated longitudinal depthward acoustic waves (DAW) contributes to overcoming the substrate adhesion, while the propagation of transverse surface acoustic waves (SAW) accounts for the horizontal movement of the nanosheet. Nevertheless, there are still some technical points needed to be addressed:

1. The authors investigated the light-actuated moving speed of the nanosheet as functions of the laser repetition frequency and the pulse energy, given in Fig.3. It will be nice to understand the relationship between moving speed and the thickness of the nanosheet for better understanding the physical mechanism.

Reply: Many thanks for your positive feedback and helpful comments.

Considering your reasonable suggestion, we tried to carry out some experiments about the relationship between the moving speed and thickness. Unfortunately, we could not find it. The possible reasons are: (1) The size and shape of nanosheets prepared by mechanical exfoliation are random, it is almost impossible to obtain a series of nanosheets with same morphology but different thicknesses, but in our calculations, the morphology is an important index that can determine the van der Waals adhesion and the thermal fields. (2) It is practically difficult to identify the nanosheets that have been moved before by the laser in the atomic force microscope. (3) According to our new concept presented in the revised manuscript, the motion of nanosheets is induced by their local overheating occurring due to gaps between VSe₂ nanosheets and substrates. The location and the thickness of these gaps are uncontrollable in the present preparation method, but still determining the thermal stress and thus the moving speed.

Therefore, such a dependence is not clear due to the complexity of the phenomenon. In our calculations, the driving force is the thermal stress inside the nanosheet. We calculated the heating temperature by laser radiation of nanosheets with different thicknesses, as shown in Fig. R1a by solid curves. A more complete dependence is shown in Fig. 3b in the revised manuscript. We

obtain the maximum of the average nanosheet temperature corresponding to the thicknesses of 50 nm. However, it does not consider the change in the rate of electron-phonon relaxation with the thickness of the nanosheet, which also affects the temperature. In addition, we calculated the dependence between vertical temperature pressure and thickness, as shown in Fig. R1a by dashed curves and in Fig. 3c in the revised manuscript. It shows a monotonous increment of the vertical temperature pressure with the thickness of the nanosheet, which greatly complicates the theoretical consideration of thermoelastic waves.

Therefore, at the moment we cannot report an explicit relation between the speed of the nanosheet's movement and its thickness. We hope that this issue will be clarified with further studies. What can be said now is that too thin nanosheets (several nm thick) will hardly heat up enough to cause the thermoelastic stress necessary for movement.

In the revised manuscript, these reasonings are reflected in the DISCUSSION part, especially in "3.1 Nanosheet thermal analysis" subsection. See also our reply to the comment 2 of Review 1.

2. The authors mainly compared the manipulation behavior between VSe₂ and MoS₂ nanosheets. How about other 2D materials? More discussions are expected to explore the material dependency.

Reply: We understand the importance of this question, which permeates all the reviews. So, we performed additional experimental research. We utilized mechanical exfoliation method to prepare several 2D materials nanosheets on the same sapphire substrates, including h-BN, MoS₂, WSe₂, PdSe₂, TiSe₂, as well as VSe₂. Then we used the same optical drive technique to actuate them. The moving results are listed in Table R1. Only TiSe₂ performs similar behavior with VSe₂, other 2D materials cannot move under laser irradiation. The new experimental data also showed better indicators for the speed efficiency of the movement of the VSe₂ nanosheets.

Table R1. Manifestation of the optical driving in different nanosheet/substrates systems.

Nanosheet-Substrate System	Movement Results	Speed ($\mu\text{m/s}$) / Energy (nJ)	Efficiency of Movement ($\mu\text{m}\cdot\text{s}^{-1}\cdot\text{mW}^{-1}$)
h-BN-Sapphire	No	-	-
MoS ₂ -Sapphire	No	-	-

WSe ₂ -Sapphire	No	-	-
PdSe ₂ -Sapphire	No	-	-
TiSe ₂ -Sapphire	Yes	0.70 / 11.0	64
VSe ₂ -Sapphire	Yes	1.83 / 5.0	366

Then we analyzed the optical, thermal, and acoustic properties of these 2D materials (Table R2), which can be potential parameters related to the motion mechanism. They include the absorption coefficient α , reflectivity R , linear thermal expansion coefficient α_{TE} , thermal conductivity κ , specific heat C_p , density ρ , average (upper bounds polycrystals) values of bulk modulus \bar{B} , Young's modulus \bar{E} , and Poisson's ratio $\bar{\nu}$.

Table R2. Optical, thermal, and acoustic properties of materials (The original references are given Supplementary Note 5 in Supplementary Information).

Material	$\alpha_{E_{1c}}$ (cm ⁻¹) at 1040 nm	$R_{E_{1c}}$ at 1040 nm	α_{TE} (10 ⁻⁶ K ⁻¹)	κ (Wm ⁻¹ K ⁻¹)	C_p (JK ⁻¹ kg ⁻¹)	ρ (g cm ⁻³)	\bar{B} (GPa)	\bar{E} (GPa)	$\bar{\nu}$
2D materials									
hBN	<9.5	nonissue	38.0 (c -axis) 2.73 (a -axis)	2 (c -axis) 400 (a -axis)	794	2.18	196.7	344.1	0.21
MoS ₂	4.8·10 ³	0.40	8.65 (c -axis) 1.9 (a -axis)	2.0 (c -axis) 85 (a -axis)	397	5.06	49.6	83.3	0.22
PdSe ₂	2.0·10 ⁴ (7L)	0.37	no data found	1.51 (c -axis) 28.5 (12L, a -axis)	266	5.94	36.6	60.1	0.23
TiSe ₂	3.3·10⁵ (exp.) 5.8·10⁵ (theory)	0.47	19.2 (c -axis) 18.4 (a -axis)	1.85 (c -axis) ≈20 (a -axis)	345	4.69	27.0	46.6	0.21
VSe ₂	3.3·10⁵ (exp.) 7.8·10⁵ (theory)	0.48 0.40	9.8 (c -axis) 18.6 (a -axis)	2.99 (c -axis) 7.32 (a -axis)	369	5.825	27.1	50.4	0.19
WSe ₂	4.0·10 ⁴ (5L)	0.41	16.7 (c -axis) 11.1 (a -axis)	1.5 (c -axis)	212	9.2	41.8	76.6	0.19
Substrates									
α -Al ₂ O ₃ (sapphire)	~0	nonissue	7.3 (c -axis) 8.1 (a -axis)	25.8 (c -axis) 23.0 (a -axis)	775	3.97	231.8	368.7	0.23
Si	21.7		3.6	148	712	2.33	83.3	151.3	0.20
JGS3 quartz glass	~0		0.55	1.4	670	2.20	37	72	0.17

The conclusion is obviously that VSe₂ and TiSe₂ are distinguished by a huge value of the absorption coefficient, which indicates the thermal nature of the movement. The second difference

lies in the relatively large values of the thermal expansion coefficient, which is twice as high as that of sapphire. Thus, our observations confirm the thermoelastic mechanism of the movement of nanosheets. Under the condition of inhomogeneous irradiation of pulsed laser, the nanosheet with larger absorption coefficient can acquire a larger temperature gradient. A larger gradient can lead to a stronger in-plane thermal stress, which determines non-uniform stretching and thus acoustic waves in the nanosheet plane. Due to a large thermal expansion coefficient, the resulting displacement of material in these waves reaches the value that is comparable with the nanosheet path after one laser pulse, which we have shown in the revised manuscript.

In the revised manuscript, the results of these additional studies have been added in “3.1 Nanosheet thermal analysis” subsection and Supplementary Note 5.

3. As the acoustic reflection coefficient of the nanosheet-substrate interface plays a key role in light-actuation. What will happen on different substrates?

Reply: We thank the reviewer again to pointing out this.

Similarly to the previous point, we used mechanical exfoliation method to prepare VSe₂ nanosheets on three different substrates: sapphire, quartz glass, and silicon. The moving results are listed in Table R3. We can see that VSe₂ nanosheet moves on the sapphire and quartz substrates, but stands still on the silicon substrate. The efficiencies of movement calculated turned out 434 μm·s⁻¹·mW⁻¹ on quartz substrate and 366 μm·s⁻¹·mW⁻¹ on sapphire substrate. That is, the efficiency ranking is: quartz > sapphire > silicon.

The separation distances d calculated based on the hard sphere model (see Fig. R4) are 0.372 nm, 0.255 nm, and 0.284 nm for VSe₂/quartz, VSe₂/sapphire, and VSe₂/silicon contacts, respectively. Using these values, the vdW forces per unit area (vdW pressure) has been calculated:

$$\frac{F_{vdW}}{S} = -\frac{A}{6\pi d^3},$$

where the Hamaker constant A has been found in the same formula as for VSe₂ on sapphire:

$$A = \frac{3h}{8\sqrt{2}} \frac{(n_s^2 - 1)v_p v_s}{\sqrt{(n_s^2 + 1)}[\sqrt{(n_s^2 + 1)}v_s + v_p]}$$

With the refractive index at high frequencies n_s , and the main electronic absorption frequency ν_s values taken for Si and SiO₂ [Table 13.2 in Ref. R7], we obtain $A = 0.57$ eV and $A = 0.27$ eV for the VSe₂/silicon and VSe₂/quartz contacts, respectively.

Thus, we can obtain the value of van der Waals pressure, as listed in Table R3. It reveals a correlation between van der Waals interaction and speed efficiency: the smaller the vdW adhesion force, the greater the efficiency of the movement.

Table R3. Manifestation of the optical driving in different substrates.

Nanosheet-Substrate System	Speed ($\mu\text{m}\cdot\text{s}^{-1}$) / Energy (nJ)	Efficiency of Movement ($\mu\text{m}\cdot\text{s}^{-1}\cdot\text{mW}^{-1}$)	vdW Pressure (GPa)
VSe ₂ -Quartz	2.78 / 6.4	434	0.044
VSe ₂ -Sapphire	1.83 / 5.0	366	0.199
VSe ₂ -Silicon	Motionless	0	0.211

The results appeared in the “1. Van der Waals adhesion and friction forces” subsection of the revised manuscript. However, the role of acoustic reflection in our new version remains uncovered. The fact is that, as we mentioned in the reply to comment 2 of Review 1, the vertical thermal stress monotonously increases with thickness, but for sufficiently thin nanosheets (10-20 nm) it is insignificant compared to the possible friction force. Therefore, in the simplistic consideration of the longitudinal elastic wave, which we limited in our work, we did not consider it. We hope that further studies will fill this gap.

4. Is the nanosheet suspended above the substrate or still weakly adhered to the substrate after light-actuation? If the nanosheet is adhered to the substrate, what’s the magnitude of the friction force between the nanosheet and the substrate?

Reply: We thank the reviewer for this important comment.

In our previous approach, it was assumed that the nanoparticle is lifted by vertical pressure, and stands on Rayleigh waves like a car on wheels and rides on the surface. In this regard, friction was implied only by the rolling friction, and was not considered due to insignificance. Now, after the

calculations of the temperature field, we understand that vertical stress is hardly sufficient to overcome the van der Waals forces (at least for nanosheets with the thickness of 10-20 nm). Besides, the notion about the Rayleigh character of surface waves was not confirmed. Therefore, in the revised manuscript, we changed the concept and came to the conclusion about propagating thermoelastic Lamb waves. In this condition, the nanosheet/substrate contact remains to a significant extent, and thus the nanosheet really has to overcome the friction force.

We considered the friction force within the hard-sphere model which suggests [Ch. 18 in Ref. R7]:

$$F_{fr} = 2\varepsilon S \Delta\gamma / \delta,$$

where $\Delta\gamma$ is the part of the adhesion surface energy consumed when the spheres of the top surface detach from those of the bottom surface in order to go over them, δ is the horizontal displacement of a sphere in one cycle of this movement, and ε is the part of the kinetic energy of the moving sphere transferred to the bottom surface. Generally, $\Delta\gamma = \Delta W_{adh}/2$, the half of adhesion work spent by a sphere of the upper surface to move between the states with lowest and highest energies.

The problem of finding ΔW_{adh} is similar to that given in the Example 18.2 of Ref. R7 and has been considered similarly, however, our consideration was based on obtaining a maximum estimate of the friction force. In our case, at the contact plane, selenium atoms from the top VSe₂ side and oxygen atoms from the bottom sapphire side contribute to the friction force. Oxygen atoms form in the contact plane are organized into equilateral triangles with a side equal to $2r_O = 0.25215$ nm.^{R3} The maximal ΔW_{adh} will be achieved when selenium atom moves through the neighboring triangle along its mean line. At that, the atom passes two different loading-unloading adhesion cycles, as shown in Fig. R5. In the first cycle, selenium passes through the middle of the side of the triangle, and in the second through its vertex. Based on the geometry shown in the figure and the hard-sphere radii known from crystallographic data, it is easy to obtain: $\Delta\gamma = \Delta\gamma_1 + \Delta\gamma_2 \approx -0.356 W_{adh}/2$ and $\delta = \delta_1 + \delta_2 = 2\sqrt{3}r_O$, where indices 1 and 2 correspond to the first and second cycles, respectively. The detailed math is given in Supplementary Note 4.

Fig. R5 The path of the selenium atoms moving along the contact oxygen lattice of sapphire. The yellow and blue balls represent selenium atoms and oxygen atoms.

For an atom of the top surface contacting three atoms of oxygen in sapphire, an upper bound approximation is known: $W_{adh}^+ \approx \sqrt{3}A/4\pi^2r_0^2$.^{R6} Therefore, for an ideal contact when all selenium atoms lie in the equilateral triangles of oxygen, the maximal ($\varepsilon = 1$) shear stress due to friction for VSe_2 will be:

$$\sigma_{fr} = \frac{F_{fr}}{S} = \frac{C_{\Delta\gamma}A}{8\pi^2r_0^3},$$

with a coefficient $C_{\Delta\gamma} = -0.356$. Since we had the Hamaker constant calculated in the main text ($A = 0.39$ eV), we obtain $\sigma_{fr} = -0.141$ GPa. For the VSe_2 nanosheet with the scale of $3.7 \times 3.1 \mu\text{m}^2$, the maximal friction force is therefore $F_{fr} = -1.62$ mN.

According to our new thermal calculations, the horizontal thermal stress caused by possible gaps between the nanosheet and the substrate (Fig. R1b) can exceed the maximum friction stress by an order of magnitude. Therefore, our optical drive method can effectively overcome the friction force and actuate the nanosheet.

In the revised manuscript, we added the discussion of friction force in “1. Van der Waals adhesion and friction forces” subsection and Supplementary Note 4.

Review #3

This manuscript reports an optical manipulation method to transport single VSe₂ nanosheets on a polished sapphire substrate in air. Considering that most of the previous studies on optical manipulation of micro-/nanostructures were done in liquid and few optical methods could overcome the Van Der Waals interactions, this work shows a good level of novelty and provides a new method to the optical manipulation community based on the photoacoustic effects. The major weakness of this work, however, is that this method seems to be limited to a specific system (i.e., VSe₂ nanosheets on a sapphire substrate) and cannot be generalized to other nanostructures (e.g., MoS₂ nanosheets could not be moved in this work). If the authors can demonstrate or provide ways to generalize this method, this work will certainly be more attractive.

Reply: We are very grateful to the reviewer for the positive assessment of our work and the valuable comments.

To generalize the method, we carried out additional studies of several materials (as we already wrote in details above when replying to Review 1, and the comment 2 of Review 2). Sensitivity to the optical drive was shown by VSe₂ and TiSe₂ nanosheets, both semimetals with close large values of the absorption and the thermal expansion coefficients. These results allow us to predict the applicability of our method to semimetals, and also additionally testify in favor of the thermoelastic mechanism of motion.

The methods discovered before (such as plasmonic-driven gold nanoplates and others) are also limited by specific material properties. But together they form a tool set applicable in different tasks of the micro- and nano- manipulations.

Another weakness is that the manuscript attributed the mechanism to the photoacoustic force after excluding other possible mechanisms. However, it lacks direct experimental evidence of the existence of the photoacoustic force and characterization of its features. The discussions were based on literature, but the conditions in the literature were not necessarily the same. The authors can consider some pump-probe experiments to make the photoacoustic mechanism more convincing. Note that the acoustic vibrations of nanostructures on substrates have been studied by ultrafast pump-probe spectroscopy, e.g., 1. Yu, K.; et al., Damping of Acoustic Vibrations of Immobilized Single Gold Nanorods in Different Environments. Nano Letters 2013, 13 (6), 2710-

2716. 2. Wang, J.; et al., Long Lifetime and Coupling of Acoustic Vibrations of Gold Nanoplates on Unsupported Thin Films. The journal of physical chemistry. A 2019, 123 (47), 10339-10346.

Reply: We thank the reviewer for this suggestion which will help us to improve the quality of this work. To fulfill the data, we performed micro-pump-probe experiments with VSe₂ nanosheets.

The detailed illustration of our home-built pump-probe technique is given in METHODS section of the main text and in Supplementary Note 6 of Supplementary Information. Using this system, we studied the ultrafast dynamic process of VSe₂ nanosheet, as shown in Fig. R6a. Some oscillations are found in the relaxation process. We utilized a bi-exponential model to fit the curve: $\frac{\Delta R}{R} = A_1 \exp\left(-\frac{\tau}{\tau_1}\right) + A_2 \exp\left(-\frac{\tau}{\tau_2}\right)$, where A_1/A_2 and τ_1/τ_2 represent the amplitude and lifetime of the fast/slow relaxation process. The fitted fast and slow relaxation lifetime is $\tau_1 = 85.1$ ps and $\tau_2 = 599.5$ ps with the amplitudes $A_1 = 0.516$ and $A_2 = 0.944$. Thank the reviewer's suggestion, in the revised manuscript, we added the experimental results in Supplementary Note 6 and modified our theoretical calculations of heating dynamics based on our experimental data.

Fig. R6 Ultrafast dynamics processes of VSe₂ nanosheets on sapphire. **a** A bi-exponential fitting curve is shown by the red curve. **b** A Fourier spectrum of the difference between the experimental and fitting signals.

On the other hand, the frequencies of the oscillations in the relaxation curve are observed in the Fourier spectrum of the remainder (Fig. R6b). In addition to high-frequency noises, we obtain the

oscillation with a frequency of ~ 14 GHz due to coherent acoustic phonons (CAP) signals, which may be the feature of the surface waves. However, we cannot simply interpret them as prove of the photoacoustic mechanism. The CAP signals are weak and mainly observed when pumped at 520 nm rather than at 1040 nm, which corresponds to a different mechanism of their generation from our optical drive. In addition, the frequencies of the thermoelastic waves that we used to explain the movement of the nanosheet may be less than 1 GHz, and, unfortunately, this region is not accessible by our current methods. Therefore, we rely more on indirect signs and analysis of the physical properties of materials, which we have significantly expanded in the revised manuscript.

We believe that the interest in the problem generated by our communication will encourage further research, which will reveal more explicit evidence of the generation of surface waves in the 2D nanosheets.

Some other comments are listed below.

1. The abstract stated that “The speed efficiency ... exceeds that of optical trapping by three orders of magnitude.” This is not accurate. The conclusion is based on results in Ref. 50, but that work is not the normal “optical trapping” but rather a special experiment that used propagating surface plasmons to transport nanoparticles. Traditional optical traps can already move nanostructures with high speeds, and the modern phase gradient optical traps can drive nanostructures even faster.

Reply: Thank the reviewer for the correction.

In the revised manuscript, we removed the corresponding sentences from Abstract and Conclusion.

2. The manuscript used a contact distance $d=0.20\text{nm} \pm 0.05\text{nm}$ for calculations based on the results in Ref. [26]. That paper studied VSe₂ thin films synthesized by molecular beam epitaxy on sapphire substrates. The contact distance of exfoliated nanosheets on a polished sapphire substrate in this work is likely much larger than that, especially considering the roughness of the substrate and the result shown in Fig. 1d.

Reply: Thanks for drawing attention to this aspect.

(1) First of all, we agree with the reviewer that the average contact distance between nanosheet and substrate by mechanical exfoliation method is apparently larger than that by molecular beam epitaxy due to the possible island-type of adhesion and the absence of thermal expansion of the VSe₂ lattice during nano-deposition. However, the exact distance value cannot be measured with the available experimental equipment. Therefore, we must confine ourselves to overestimates, but ensure that they are as reasonable as possible. To do this, we introduced the rigid sphere model to estimate the contact distance d .^{R7} The contact distance d can be regarded as the distance between centers of the lowest spheres of the top surface and the highest spheres of the bottom (see Fig. R4). The hard sphere radii of the surface atoms (oxygen from the sapphire side and selenium from VSe₂ side) are taken as half minimal O – O and Se – Se distances in the contacting planes, respectively: $r_{\text{O}} = 0.126 \text{ nm}$ ^{R3} and $r_{\text{Se}} = a_{\text{lat}}/2 = 0.168 \text{ nm}$ ^{R4} (where a_{lat} is the crystal lattice parameter). Oxygen atoms in the contact surface of *c*-sapphire form a face of hexagonal close-packed lattice,^{R5} and with a tight package, atoms of the top surface should lie in hollows of equilateral triangles formed by oxygen atoms.^{R6} Then it is easy to find $d = \sqrt{(r_{\text{O}} + r_{\text{Se}})^2 - 4r_{\text{O}}^2/3} = 0.255 \text{ nm}$. Surprisingly, it turned out to be close to the magnitude observed in the TEM image ($d \approx 0.2 \text{ nm}$) of Ref. 26 (now Ref. 32).

However, we have to mention that this assessment is suitable for the case of ideal contact, which apparently takes place when the nanosheet is prepared by molecular beam epitaxy. In the case of mechanical exfoliation method, a less optimal relative arrangement of the lattices of the top and bottom surfaces exists. Here, we limit ourselves to the upper bound estimate and take d as 0.255 nm in the revised manuscript. We have tried our best to decrease the errors, and we believe that it can be evident enough for understanding the order of magnitude of van der Waals interactions.

In the revised manuscript, we added these calculations in “1. Van der Waals adhesion and friction forces” subsection.

(2) Another issue is the surface quality of the substrate. We are sorry that our illustration of the nanosheet surface caused a misunderstanding about the substrate surface. The large oscillations seen in the Fig. 1d on the substrate side were represented by the instrumental noise. Of course, we had to provide a better analysis of the roughness of the substrate surface, but we relied on the values given in the manufacturer's certificate ($R_a < 0.2 \text{ nm}$). Now we have fixed this flaw. We examined the surface quality of the sapphire substrate with a more sensitive instrument (Veeco

Instruments MultiMode V), as shown in Fig. R3. It can be found that its roughness is indeed within 0.2 nm, which confirms the validity of using contact distances exceeding this value.

In the revised manuscript, we added the roughness analysis in Supplementary Note 3.

3. The manuscript said, “the Raman spectrum from 100 to 300 cm^{-1} under 532 nm excitation is presented in Fig. 2c” in the DISCUSSIONS but “Raman characterization was ... excited by 633 nm cw laser.” in METHODS. Which wavelength is correct?

Reply: Many thanks to pointing out this mistake.

The wavelength was 532 nm, we have corrected the mistake in the revised manuscript.

4. In Fig. 2, the laser spot of 5 μm radius should cover all the four nanosheets in the blue square, why was only the middle one moved?

Reply: The comment is insightful, we really appreciate it.

Firstly, upon careful examination of Fig. 2, you might see that the right nanosheet also moves a little. We added two border lines in the Fig. R7, which may help you to see the motion more clearly.

Fig. R7 Optical drive for VSe₂ nanosheets at the frequency of 1 Hz. The white lines and arrows show the movement of the right nanosheet.

The effect of the preferable motion of the upper nanosheet is easily explained in the new concept that we present in the revised manuscript: being partially superimposed on the others by the edges,

it has large gaps with the substrate. The resulting local overheating creates stress that moves this nanosheet. At the same time, the heating of the remaining nanosheets is not large enough since they are attached tightly to the substrate.

In the revised manuscript, we added the discussion in “3.3 Photophoretic force” subsection.

5. How did the authors measure the results in Fig. 3? Did they move the laser spot? The maximum distance was larger than 10 μm so it is likely that the laser spot was moved. How would that affect the measurement? In addition, if the laser spot was moved, optical trapping force will play a role in the movement of the nanosheet because a large intensity gradient exists at the edge of the pattern even for a flat-top beam.

Reply: Thank the reviewer for this important remark.

(1) Indeed, the laser spot was moved after a nanosheet passed through it. We took the video frame by frame, and measured the moving distance and moving time of the nanosheet when the laser was absolutely still, and thus we obtained the moving speed of nanosheet. Therefore, the experimental data in the left side of Fig. 3 (now Fig. 2) did not contain the time information of moving the laser. On the other hand, we tried to draw its trajectories to make the moving process more visual, we deleted the time length when the laser was moving and obtained the moving traces at the time scale of approximately 10 s. Therefore, we believe that this approach should form a correct representation of the movement of the nanosheet.

(2) We thank the reviewer for reminding the edge gradient. In our case we have a uniform top-hat beam, but at the boundaries of the beam spot, the intensity should have a gradient. This region is schematically shown by a ring in Fig. R8. Based on diffraction constraints, the width of this region should be on the order of the wavelength.

Fig. R8. Scheme of the gradient field region on a VSe₂ nanosheet when the laser beam spot is moved.

Therefore, at intensities of 10^{15} W/cm², the gradient ∇I can reach 10^{21} W/m³. The force exerted on an electric dipole μ by the light field E :^{R8}

$$\mathbf{F}_{grad} = (\boldsymbol{\mu} \cdot \nabla) \mathbf{E} = \frac{\alpha_d \nabla E^2}{2} = \frac{\alpha_d \nabla I}{2\epsilon_0 c},$$

where α_d is the dipole polarizability. Here we neglect the refractive index of the environment which is air in our case with $n_{en} \approx 1$. If we consider the total dipole moment induced in a streak which is the gradient area in the nanoparticles in Fig. R8, its polarizability can be obtained from the Lorentz-Lorenz equation:

$$\alpha_d = 3\epsilon_0 V \frac{n^2 - 1}{n^2 + 2},$$

with the volume $V \approx aa_x L$, where $a = 3.7$ μm is the length of the nanosheet, $a_x = 1$ μm is the width of the gradient region and L is the nanosheet thickness. Taking into account a known value of the refractive index of VSe₂: $n(1040 \text{ nm}) \approx 3$,^{R9} we obtain for the gradient field:

$$F_{grad} = \frac{12aa_x L}{11c} \nabla I,$$

that gives approximately 127.8 nN. The point-dipole approximation applied here is formally valid for nanoparticle sizes $a < \lambda/20$. However, it has been previously shown that the gradient optical force calculated within this approximation is less sensitive to the particle size, so that even at $a \sim \lambda$ the result differs from that of the more rigorous Generalized Lorenz-Mie theory by no more than one order of magnitude.^{R8} Therefore, we can exclude the gradient force from consideration.

For a complete overview of probable forces, we added this estimate in “2. Optical force” subsection.

6. A major question is how the laser controls the moving direction of the nanosheet? If the laser fully covers a nanosheet, why is the acoustic wave directional? The nanosheet in Fig. 5 apparently moved in a curve, how would that happen?

Reply: Thanks for drawing attention to this issue.

Indeed, this is a fundamental question that puzzles when observing the effect. In our previous manuscript, we did not give an answer to it, believing that the direction of movement occurs randomly. Now, after additional research, we promote a hypothesis that the cause of the movement is horizontal thermal stress that occurs in the nanosheet due to a bad contact to the substrate in certain places (presumable, at an edge), as seen in Figs. R1b and R2, and in more details in the revised manuscript. In this case, the direction of movement is determined by the location of these gaps under the nanosheet. Unfortunately, we still have no explicit experimental evidence for this, but the analysis of temperature fields, which we present in the revised manuscript, suggests such a conclusion. The movement along the curve can also be understood according to this assumption: when moving, the nanosheet can run into new surface defects, and the areas of heat release on it can thus shift.

In the revised manuscript, we added a detailed simulations of the thermal stress and discussion of the gaps at the interfaces in “3.1 Nanosheet thermal analysis” and “3.2 Thermoelastic effect” subsections.

7. How was the “escape momentum of the vdW force” calculated?

Reply: As a reminder, the escape momentum is generally found as that required for overcoming a binding potential U , so the momentum is $\sqrt{2m|U|}$. The potential energy of adhesion between two vdW surfaces is: $U_{vdW} = -AS/12\pi d^2$, where m and S are the mass and the area of a nanosheet, A is the Hamaker constant, and d is the separation distance between the nanosheet and the substrate.

However, in the revised manuscript, the vertical stress is considered insignificant in many cases, so we do not need this estimate, which is now removed.

8. The manuscript said, “the initial offset energy derived from the horizontal intercept in Fig. 3b is approximately 7.8 nJ”. Why didn’t the authors experimentally demonstrate that by tuning the laser pulse energy to identify the threshold?

Reply: This is another insightful comment that we appreciate very much.

Unfortunately, the instrument sensitivity does not allow to prescribe the threshold area more subtly than it was done in Fig. 3b (now Fig. 2b). However, our observations and the view of the dotted curve indicate that there is a small step at ~ 9 nJ, all the points below 9 nJ are zero, as shown in Fig. R9. It seems to us that this observation is valuable because it reveals the nature of the adhesive friction forces consisting of static and sliding friction forces. The former is necessary to set the nanosheet in motion, and the latter is less than the former and is spent on maintaining the movement [Ch. 18 in Ref. R7]. When the laser energy is small, it is all spent on overcoming the static friction force, and when the energy noticeably exceeds this threshold force, then most of it is spent on overcoming the sliding friction force. Thus, by measuring speed-energy characteristic, we can test the static/sliding ratio of the vdW friction force. The static friction force corresponds to the step position (*i.e.* 9 nJ), while the sliding friction force corresponds to the E -intercept of the linear approximation (7.8 nJ).

In the revised manuscript, we added the above discussions in the “Results” section.

Fig. R9 The relationship between light-actuated moving speed of the nanosheet at $f_{rep} = 1$ kHz and the laser pulse energy.

References

- R1. Pierce, A. D. *Acoustics: An introduction to its physical principles and applications*, (Springer Nature Switzerland AG, 2019).
- R2. Ruello, P. & Gusev, V. E. Physical mechanisms of coherent acoustic phonons generation by ultrafast laser action. *Ultrasonics* **56**, 21 (2015).
- R3. Maslen, E. N., Streltsov, V. A., & Streltsova, N. R. Synchrotron X-ray Study of the Electron Density in α -Al₂O₃. *Acta Cryst. B* **49**, 973 (1993).
- R4. De Jong, M., *et al.* Charting the complete elastic properties of inorganic crystalline compounds. *Scientific Data* **2**, 150009 (2015).
- R5. Zhang, D. *et al.* Strain Engineering a $4a \times \sqrt{3}a$ Charge Density Wave Phase in Transition Metal Dichalcogenide 1T-VSe₂. *Phys. Rev. Mater.* **1**, 024005 (2017).
- R6. Lipkin, D. M., Israelachvili, J. N. & Clarke, D. R. Estimating the metal ceramic van der Waals adhesion energy. *Philosophical Magazine A* **76**, 715 (1997).
- R7. Israelachvili, J. N. *Intermolecular and Surface Forces*. (New York, 1992)
- R8. Harada, Y. & Asakura, T. Radiation forces on a dielectric sphere in the Rayleigh scattering regime. *Opt. Commun.* **124**, 529 (1996).
- R9. Rugut, E. K. Numerical simulation of structural, electronic and optical properties of transition metal chalcogenides. (Univ. of the Witwatersrand, Johannesburg, 2017).
- R10. Neuman, K. C. & Block, S. M. Optical trapping. *Rev. Sci. Instrum.* **75**, 2787 (2004).

List of changes in the main text of the revised manuscript:

1. In Abstract part:

The sentences “The speed of this movement exhibits a linear growth with increasing pulse energy and repetition frequency. The actuator is powered by the combined action of thermally generated depthward and surface acoustic waves in ultrathin VSe₂ nanosheets. The former builds an uplifting force on the nanosheets to overcome the adhesion force of ~ 4.7 mN, while the latter provides the lateral momentum to actuate the nanosheets. The speed efficiency of our actuator is $65.5 \mu\text{m}\cdot\text{s}^{-1}\cdot\text{mW}^{-1}$, which exceeds that of optical trapping by three orders of magnitude.” are changed to “The driving force can be explained by a photoacoustic mechanism originated from nanosheet edge overheating, which can generate a significant thermal stress and corresponding thermoelastic waves. Through the analysis of different 2D material-substrate systems, a large absorption coefficient ($\sim 10^5 \text{ cm}^{-1}$) is the crucial property for this optical drive. For such materials, the thermal expansion coefficient and the bulk modulus become significant. The maximal speed efficiency observed in the VSe₂ actuator is $434 \mu\text{m}\cdot\text{s}^{-1}\cdot\text{mW}^{-1}$ in our experiment, which is much higher than efficiencies of previous reported photoacoustic manipulations.”.

2. In Introduction part:

The sentences “We analyze the forces generated in the nanosheet and explain its movement from the standpoint of photoacoustic effects: the ultrafast generation of longitudinal depthward acoustic waves (DAW) contributes to overcoming the substrate adhesion, while the subsequent generation and propagation of transverse surface acoustic waves (SAW) are responsible for the horizontal movement of the nanosheet.” are changed to “Furthermore, we investigate the properties of various 2D materials to find the sensitive parameters in the optical drive. We analyze its possible mechanisms in terms of the magnitudes of the forces acting in them and present our understanding of the nature of the discovered phenomenon. The latest related research reported the laser-driven complex motion of gold nanoplates on fixed microfibers, where photoacoustic Lamb waves overcome friction at the plane-cylinder contact line and make the nanoplates rotate around the fiber and slide along it. While their optical drive takes advantages of the surface plasmon polaritons excited in the metal nanoplates, our optical drive utilized the thermoelastic

effect in the plasmon-free geometry. At the same time, our optical drive provides 2D planar motion, that was not observed in these works.”.

3. In Results part:

The sentences “In our experiment, the pulse mode of irradiation is a crucial factor for driving the nanosheets. When we switch the driving source to a continuous-wave (CW) laser (the wavelength $\lambda = 1064$ nm) or to a quasi-continuous laser (the repetition frequency $f_{rep} = 80$ MHz, $\lambda = 800$ nm), we do not observe the motion of the nanosheet.” are added.

The sentences “The speed of motion increased quasi-linearly with increasing laser energy.” are changed to “It has a step-like threshold, and then increases linearly with laser energy. We associate this behavior with the manifestation of the dual character of adhesive friction forces consisting of static and sliding friction. When the laser energy is small, it is all spent on overcoming the static friction force, and when the energy noticeably exceeds this threshold force, then most of it is spent on overcoming the sliding friction force. Thus, by measuring speed-energy characteristic, we can test the static/sliding ratio of the vdW friction force. The static friction force corresponds to the step position (i.e. 9 nJ in Fig. 2b), while the sliding friction force corresponds to the E-intercept of the linear approximation (7.8 nJ).”.

4. In Discussions part:

It has been reorganized representing four subsections: “1. Van der Waals adhesion and friction forces”, “2. Optical force”, “3. Photoacoustic mechanisms” (“3.1 Nanosheet thermal analysis”, “3.2 Thermoelastic effect”, “3.3 Photophoretic force”) and “4. All-optical splicing of 2D nanosheets”. Subsections 3.1 and 3.2 are completely new, but others have undergone revision and partial changes. The changes are marked in GREEN in the main text.

5. In Conclusion part:

The sentences “The traditional optical force and photoacoustic force were ruled out in our experiment, and we proposed a photoacoustic mechanism to explain the motion of 2D VSe₂ nanosheets, including DAW and SAW. The theoretical calculation demonstrated that the DAW-

induced force could assist the nanosheet in balancing the vdW force, and the SAW-induced force was responsible for the translational motion.” are changed to “The nanosheets can be also moved on quartz substrate with a lower adhesive force, and cannot be moved on silicon substrate with a larger adhesive force. The maximal speed efficiency of nanosheet manipulation achieved in our experiment is $434 \mu\text{m}\cdot\text{s}^{-1}$ per milliwatt, which is even larger than previously demonstrated in other experiments with gold nanoplates on microfibers. The moved 2D materials belong to semimetals and differ in large values of absorption and thermal expansion coefficients. 2D materials with essentially lower absorption coefficients such as BN, MoS₂, WSe₂, and PdSe₂ do not demonstrate sensitivity to the optical drive. We assign the phenomena to the thermoelastic mechanism of surface acoustic wave generation. Numerical study of the spatial and temporal distributions of the heating temperature attained upon laser irradiation reveals the possibility of the generation of an essential thermal stress in the nanosheet in the case of an air gap between the nanosheet and the substrate at the nanosheet edges. We demonstrate that the longitudinal surface waves resulting from this thermal stress carry a significant momentum that is sufficient to move the nanosheet along a horizontal surface at a speed close to that observed in the experiment. Other possible mechanisms, including optical pressure, the photoacoustic mechanisms except thermoelastic force, and photophoretic force, can be negligible in this optical drive.”.

6. In Methods part:

The sentence “The VSe₂ nanosheets was mechanical exfoliated from bulk single crystals (commercial product, 1T phase) and then transferred to a polished sapphire substrate (roughness, Ra < 0.2 nm).” is changed to “The 2D nanosheets were mechanical exfoliated from bulk single crystals and then transferred to the polished sapphire, IR quartz glass, and silicon substrates.”.

The sentence “Surface quality of the substrates was checked using Veeco Instruments MultiMode V atomic force microscope.” is added.

The word “633 nm” is changed to “532 nm”.

The sentence “The illustration of our optical drive system is shown in Supplementary Fig. 1.” is added.

The sentences “**Pump-Probe System:** The ultrafast pump-probe measurements were performed using the same laser source (1040 nm, 380 fs, 10 kHz). The pump and probe pulses were split from

the laser source by an ultrafast beam splitter. Either pump or probe beam was doubled to 520 nm, as required. The pump beam was modulated by a chopper, collinearly combined with the probe beam and delivered into the microscope. The laser beams were focused onto the sample with the same 50× objective lens. The reflected beams were passed through a filter to block the pump beam, and was then directed into a silicon detector to obtain differential-reflection signal $\Delta R/R$. The illustration of our pump-probe system is shown in Supplementary Fig. 5.” are added.

The sentences “**Computational methods:** Heat equation system has been solved by the finite difference method using an implicit four-point algorithm. The step along the spatial coordinate was chosen $dz = 0.1$ nm, and along the time $dt = 0.5$ ps. The error associated with the finite step quantity was less than 0.5%. The time decay of the averaged over nanosheet thickness temperature difference $\Delta T_{av}(t)$ was fitted by three exponents to extrapolate the temporal dependence of the thermal stress to a larger time range. The integral in Eq. (9) has been computed by the Simpson method with the step $d\tau = 1$ ps. The $(\Delta x, \Delta t)$ mesh in Figs. 4c and 4g is (13.6 nm, 100 ps).” are added.

7. In References part:

The references “*Acta Cryst. B* 49, 973 (1993)”, “*Scientific Data* 2, 150009 (2015)”, “*Rev. Sci. Instrum.* 75, 2787 (2004)”, “*Opt. Commun.* 124, 529 (1996)”, “*Nano Lett.* 22, 6509 (2022)”, “*Ultrasonics* 56, 21 (2015)”, “*Phys. Chem. Chem. Phys.* 21, 132 (2019)”, “*Fluid Mechanics (Course of Theoretical Physics)*. (Pergamon Press, 1987)”, and “*Theory of Elasticity (Course of Theoretical Physics)*. (Pergamon Press, 1986)” are added.

Responses to Reviewers

Reviewer #2 (Remarks to the Author):

In the revised manuscript, Chen et al. provided more detailed information on this experiment and addressed part of my concerns. It is nice to see that the technique can be applied to different materials. However, I still have some concerns on this work:

1. In pages 5-6, the authors claimed that continuous-wave laser cannot drive the motion of the nanosheet but did not explain in detail. In my opinion, CW laser may also induce the overheating in the edge of nanosheet to drive its motion if the mechanism is correct. I expect more discussions here.

Reply: We thank the reviewer for drawing attention to this issue, which was left without proper coverage in our manuscript.

Heating by a CW radiation can indeed be significant. In the steady state condition, the left side of Eq. (5) in our manuscript turns to zero, and the equation, after integration over z , becomes:

$$\kappa \nabla T = I(1 - R)(1 - e^{-\alpha L}),$$

where the same boundary conditions should be applied. All denominations are the same as in our manuscript, and I stands for the average radiation intensity. Using the Newton's law of cooling: $\kappa \nabla T = h \Delta T$ in the one-dimensional case, where h is the heat transfer coefficient, we can easily estimate the heating ΔT . On a long-time scale and in the air ambience the leading mechanism of heat transfer is convection. In the free convection condition, the maximal value of $h = 25 \text{ W} \cdot \text{m}^{-2} \cdot \text{K}^{-1}$ [R1]. Therefore, using the optical parameters for VSe_2 ($R = 0.44$, $\alpha = 5.6 \times 10^5 \text{ cm}^{-1}$, $L = 10 \text{ nm}$), we can find that the heating $\Delta T = 1400 \text{ K}$ similar to that obtained with femtosecond laser is reached at intensity $I = 146 \text{ kW} \cdot \text{m}^{-2}$, which is close to that we used in our experiment.

However, unlike the case of short-pulse irradiation, here we cannot consider this heating to be local, because in the steady-state regime, thermal equilibrium is quickly achieved along the entire length a of the nanosheet and all parts of it gain the same temperature. Therefore, the in-plane thermal stress will be symmetric, and the thermoelastic mechanism will not work, like in the case shown in Fig. 4(e,f,g,h). The thermal equilibrium between the parts of the nanosheet is established $\tau_{TR}^{\parallel} \sim a^2 / \chi^{\parallel} \approx 4 \mu\text{s}$ after irradiation beginning (when $a = 3.7 \mu\text{m}$), and the maximal possible

heating of the free-standing edges for this time is $I\tau_{TR}^{\parallel}(1 - R)(1 - e^{-\alpha L})/\rho C_p L = 6.5$ K, which is not enough to produce the driving effect.

In the revised manuscript, we added a few words to clarify this conclusion in “3.2 Thermoelastic effect” subsection.

2. In page 6, what’s the pulse width of the “quasi-continuous laser”? It is inappropriate to call the laser with high repetition rate as quasi-continuous laser if its pulse width is short. The authors need to double check this term.

Reply: Thanks to pointing out this.

The remark is true, the term “quasi-continuous laser” is used in a slightly different sense in laser physics: to denote the pumping mode, which allows avoiding undesirable thermal effects. We are talking about high-frequency repetitive radiation which allows to accumulate thermal effects in the matter (the repetition period 12.5 ns is much less than the thermal relaxation time, which is several microseconds).

In the revised manuscript, we changed it to “high repetition frequency laser”.

3. The authors claimed that they cannot characterize the separation distance between the substrate and the nanosheet but provided a calculated value in the case of ideal contact. Determining the separation distance in experiments is important for the understanding of underlying mechanism. The authors may characterize it by subtracting the calculated thickness of 2D material from the height of tested sample.

Reply: We thank the reviewer for his valuable comment.

Experimental techniques that could allow us to approach the measurement of such small distances and on such small areas are indeed very limited. In particular, the proposed method does not allow to do this because the AFM accuracy and the variations in nanosheet thickness are much larger than the desired distance.

However, realizing the importance of this issue, we managed to perform a cross-section characterization of a mechanically-exfoliated VSe₂ nanosheet on sapphire by focused ion beam (FIB) and high-resolution transmission electron microscopy (HRTEM). The nanosheet was one of those that moved under the action of our femtosecond laser beam, and its thickness was approximately 50 nm. The microscopic analysis clearly revealed air gaps at the edges of the nanosheet, as shown in Fig. R1.

Fig. R1 (a) Cross-section HRTEM images of a 50-nm-thick VSe₂ nanosheet on sapphire substrate. The scale bar is 1 μm . (b, c) Enlarged images of the left (red square) and right (green square) sides in (a). The scale bar is 200 nm. (d) Enlarged image of the center side (blue square) in (a). The nanosheet thickness is about 49.59 nm. The scale bar is 50 nm. (e, f) Enlarged images of the pink square in (b) and the yellow square in (c). The scale bar is 10 nm.

At the same time, the causes of these gaps are guessed. Apparently, the reason is not in the fractures of the edges, and not in the dust on the surface of the substrate, as previously assumed, but in the nature of the separation of the nanosheet from its bulk during mechanical exfoliation:

the nanosheet is torn off in layers, and at the same time the lower layers are torn off in another place, shifted relative to the line of separation of the upper layers, and therefore become shorter than the upper ones.

The central part contacts the substrate much closer, despite the fact that the contacting layers are strongly defected, probably, as a result of the nanosheet movement. However, we can obtain the minimal contact distance of the nanosheet on sapphire, as shown in Fig. R2.

Fig. R2 Cross-section HRTEM image of the VSe₂-sapphire contact distance. The scale bar is 2 nm.

The red lines in Fig. R2 mark the VSe₂ layers, and the spacing between contacting layer and the substrate of 0.224 nm is clearly distinguishable (the uncertainty is 10-15%). It is in agreement with similar observation of the MBE VSe₂-sapphire contact by other authors in Ref. [33] and with our estimates of the contact distance by the hard-sphere model (0.255 nm).

The FFT analysis of the VSe₂ area (Fig. R3a), sapphire area (Fig. R3b), and the thin defected area between them (Fig. R3c) confirms our identification of the materials.

Fig. R3 FFT images of the (a) VSe₂ area, (b) sapphire area, and (c) defective layer on the contact area of the HRTEM image in Fig. R2.

The layer on the contacting plane is in a large extent defected, and its crystal structure is partially smoothed out, however, the VSe₂ identity can be still guessed out in Fig. R3c. Thus, the obtained results confirm our conclusions both regarding the minimum contact distance and the existence of air gaps along the edges of the nanosheet.

In the revised manuscript, we added this analysis as Supplementary Note 4, and used these results for discussion in “1. Van der Waals adhesion and friction forces” subsection of the main text.

4. In Table R2, the thermal conductivity of quartz glass is much lower than that of sapphire. Thus, the temperature of the nanosheet or the interfacial temperature difference may be significantly different on various substrates, which may influence the motion of the nanosheet. The authors may want to comment on this.

Reply: We thank the reviewer for drawing attention to this issue.

The thermal conductivity of the substrates, indeed, varies greatly, and not only in the case of quartz glass, but also, to the other side, in the case of silicon. Its influence on the heating of the nanosheet is different: while an increase in thermal conductivity (silicon) decreases temperature slightly, its decrease (quartz glass) increases it remarkably. The corresponding calculation is

shown in Fig. R4a. However, in the case of an air gap between the nanosheet and the substrate (Fig. R4b), the achieved temperature depends on the properties of the substrate very weakly.

Fig. R4 Average temperature increment of a 10-nm-thick VSe_2 nanosheet on different substrates: (a) at a tight contact with the substrate, and (b) at a 10-nm gap between the nanosheet and the substrate.

In the case of the quartz glass substrate, this temperature increment of the nanosheet at the places of a good contact versus the unchangeably high temperature at its part with an air gap leads to a decreasing thermal stress asymmetry (Fig. R5a), and finally reduces the asymmetry of the momentum of the elastic wave in the nanosheet (Fig. R5b).

Fig. R5 (a) Map of the thermal stress and (b) momentum-to-mass ratio in a 10-nm-thick VSe_2 nanosheet on the quartz glass substrate with a 10-nm edge air gap.

Within the model we applied in evaluation of the thermoelastic effect, which does not consider acoustic wave attenuation for simplicity, the wave will propagate until the thermal stress exceeds shear friction stress. Since the latter is several times less in the case of quartz glass substrate, the wave motion will be longer, and despite the lower asymmetry of the momentum, its full integral over time will be comparable with that in case of sapphire. Temporal dependence of the momentum is unchanged after first ~ 10 ns, and its integral can be characterized by the constant speed value of $v_{NS} = 2.8$ m/s. Therefore, the nanosheet displacement can be evaluated as $\Delta x \approx v_{NS} t_{end}$. Assuming that the friction force on quartz glass substrate is 4.5 times less than on sapphire substrate (according to the estimates of the vdW forces in Table 1), an extrapolation of the stress from Fig. R5a to longer times, gives $t_{end} \approx 255$ ns, and $\Delta x \approx 0.7$ nm, which is twice greater than the corresponding estimates we gave in the manuscript for VSe₂ on sapphire ($\Delta x = 0.319$ nm).

Therefore, we see that our theory explains at a semi-quantitative level the features observed in the experiment. However, its capabilities are severely limited by an insufficiently accurate knowledge of the numerous characteristics of the substance included in it. On the example of different substrates, we see that the resulting displacement nonlinearly depends on the level of friction forces. The estimates we give in our manuscript, are the upper bounds for friction, and respectively the lower bounds for displacement. Therefore, the real effect can be larger, however, its exact value cannot be theoretically obtained without precise determination of the friction forces on different substrates, which we believe can be the subject for further research.

Another notice we can make from this analysis is that the higher thermal conductivity of silicon cannot be one of the reasons of the absence of optical drive manifestation on this material, as it was mentioned in the manuscript. Now we get a better explanation for this observation: the larger level of friction nonlinearly decreases the time of thermoelastic wave motion, which makes the nanosheet motion imperceptible. It is corrected in the revised version.

In the revised manuscript, we added this analysis as Supplementary Note 10, and used these results for discussion in “3.2 Thermoelastic effect” subsection of the main text.

Reviewer #3 (Remarks to the Author):

In the revised manuscript, the authors added experimental results and theoretical analyses, some were based on my suggestions, which are appreciated. I believe the experimental results are interesting and the theoretical analyses are valuable, however, the connections between the two are weak. That is, the theoretical analyses are largely based on assumptions and indirect evidence inferred from the experimental results. They can more or less serve the role to explain the experimental observation, but the assumptions seem to lack solid basis and contain too many estimations. This is the major weakness of the manuscript.

Reply: We thank the reviewer for evaluating our work. Both the approval and critical comments of a professional are very important to us.

Realizing the importance of additional research, especially in the part concerning the morphology of the contact between the VSe₂ nanosheet and the substrate, we prepared a cross-section characterization of a 50 nm thick VSe₂ nanosheet on the sapphire substrate. The nanosheet had been pre-induced by laser to move on the substrate. High-resolution microscopy confirmed our assumptions about both the minimum contact distance and the existence of air gaps at the edges of the nanosheet. As we understand it, these were the main assumptions on which our conclusions were based.

We see that the assumptions on which our theory is based are confirmed by experiments. Of course, the theoretical description can be improved further. At each such step, however, a concrete degree of accuracy of description should be sought from the theory. In the submitted manuscript, we aimed to weed out the possible causes of the observed phenomenon and to point to the most probable one, for which sometimes rough, but, as we believe, reasonable estimates were used. We consider it necessary to inform the scientific community about our observation, especially since our theoretical vision are generally confirmed by the accumulated totality of experimental data obtained at the moment. We are confident that further confirmations, corrections or refutations of our hypotheses will more fully reveal the nature of such multifactorial phenomena, the understanding of which, of course, takes time.

Some more comments:

1. I agree with the Reviewer 1 on the concern of the calculation of vdW force, especially the 0.2 nm contact distance. On the one side, in Supplementary Note 3, the authors measured the roughness of the substrates by AFM. Those substrates should be cleaned substrates, but after transferring the nanosheets, the substrates apparently became dirty (or contaminated) as shown in Supplementary Note 7, and the roughness would be much larger. On the other side. The authors used an atomic model to get a 0.255 nm distance, that would be an ideal case and may exist only for some contact points, in that case, how do the authors choose S in the equation 1?

Reply: We are grateful to the reviewer for these important questions. We believe we can clarify them one by one.

1. Concerning the dirt, it is true that after coating, the nanosheets are surrounded in some places by other nanosheets and their fragments. However, we do not see this as a deterioration of contact. The contact of a nanosheet with the substrate was formed at the moment when the surface was absolutely clean, and in order to move the nanoparticle, it is necessary to overcome the force that was formed at that moment.

2. After the additional HRTEM study of the contact of a VSe₂ nanosheet on sapphire substrate (Figs. R1-R3), we can see that the contact is tight along the center part of the nanosheet and only has air gaps on the distance 0.2–0.5 μm from the edges (Fig R1a). The contact distance in the center part can be established by certain places where it is visible quite clearly (Fig. R2) and holds ~0.22 nm within the uncertainty of 10-20%. Some points of the contact are smoothed due to either FIB cutting procedure or previous motion of the nanosheet by laser, as shown in Fig. R6. However, no indicators of a remarkable change in the contact distance are noticed at such points. In general, the image is in a good agreement with the similar illustration in Ref. [32], which convinces us that our estimate of the minimum contact distance is correct. In addition, it should be noted that the shade of the contact layer is slightly lighter than the bulk of the substance, which is due to its increased defectiveness. However, upon closer examination, this circumstance does not affect the contact distance.

Fig. R6 An example of the partially smoothed contacting layer.

3. From the TEM images, we can see that the close contact is extended, but not pointlike. The length of this contact will certainly be different for different nanosheets, as well as in different sections of the same nanosheet. In the manuscript we mentioned an upper bound for the vdW force where we took the full area of a nanosheet. At the same time, the area is included as a multiplier in Eqs. (1) and (3) that determine the van der Waals forces and friction, respectively, so that an interested reader can always recalculate them in accordance with his idea of the most likely contact area. Concerning the thermoelastic wave consideration which we propose in our manuscript, it is not the absolute value of the vdW force is important, but the stress of its friction, i.e. the value that is independent on the contacting area.

4. Answering the direct question of the reviewer, we must apologize for the ambiguity that arose in the text and explain that the area used in the calculation of the van der Waals force was $S = 3.7 \times 3.1 \mu\text{m}^2$, the approximate area of the nanosheet shown in Fig. 1 and whose speed of motion was measured in Fig. 2. In the revised manuscript, we have rewritten this place so that there are no misunderstandings in the paragraph 5 of “1. Van der Waals adhesion and friction forces” subsection. At the same time, we tried to focus more on surface density of the vdW forces in the paragraph 3 of the same section.

2. About the optical force, note that the standard equation for gradient force is for a CW laser. For a fs pulsed laser, the equation would be different, and the authors may also need to consider

nonlinear optical interactions. In addition, the authors used a dipole model but I don't think it is appropriate to approximate the microscale sheets as dipoles. The authors said "it has been previously shown that the gradient optical force is less sensitive to the particle size, so that even at a $\sim \lambda$ it gives a result that differs from that of the more rigorous Generalized Lorenz-Mie theory by no more than one order of magnitude.³⁵" This doesn't appear to be not a solid argument to me.

Reply: Many thanks to the reviewer to bringing up this question. Let us take a closer look at it.

1. Talking about optical force, we presume the nanosheet is flat and large enough compared to the wavelength, and the light falls perpendicular to its surface. Thus, scattering forces is excluded from consideration, because a flat surface does not have curvature to scatter light except back reflection, whose momentum has been evaluated in our manuscript (P/c). However, it is the scattering forces that are the main mechanism in rare cases of observations of the optical trapping of two-dimensional nanoparticles [R2-R4]. This happens in suspensions with random orientation of nanosheets, and this is not our case.

2. Concerning transversal gradient forces, the fundamental theory points out its ponderomotive nature [R5], unrelated to light scattering. In this regard, the result from Ref. [35], indicating a weak dependence of these forces on the size of the nanoparticle, does not cause us doubts. We see a possible source of error only in an incorrect assessment of the polarizability of the microparticle (or rather, its part). To avoid this possible inaccuracy, we propose to apply a general assessment of the ponderomotive force density exerted in a bulk material with permittivity ε and mass density ρ [R6], which will be strict and suffice for our purpose:

$$\mathbf{f}_{pm} = \rho_e \mathbf{E} + \varepsilon_0 \nabla \left(\mathbf{E}^2 \rho \frac{\partial \varepsilon}{\partial \rho} \right) - \varepsilon_0 \mathbf{E}^2 \nabla \varepsilon. \quad (\text{R1})$$

Here the first term is due to free charges with density ρ_e . It turns to be zero with averaging over the light wave period. The second and third terms are dielectric ones. The Clausius-Mossotti equation gives $\rho \frac{\partial \varepsilon}{\partial \rho} = \frac{(\varepsilon-1)(\varepsilon+2)}{3}$, then

$$\varepsilon_0 \nabla \left(\mathbf{E}^2 \rho \frac{\partial \varepsilon}{\partial \rho} \right) = \frac{\varepsilon_0 (\varepsilon-1)(\varepsilon+2)}{3} \nabla \mathbf{E}^2 + \frac{\varepsilon_0 (2\varepsilon+1)}{3} \mathbf{E}^2 \nabla \varepsilon.$$

Since we consider the nanosheet's material uniform in its plane, $\nabla_x \varepsilon = \nabla_y \varepsilon = 0$. At that, thermal expansion cannot disturb it during the field application time (380 fs). Therefore, using $\mathbf{E}^2 = 2I/\varepsilon_0 c$ (here we put refractive index of the air $n = 1$), we get:

$$\mathbf{f}_{pm} = \frac{2(\varepsilon-1)(\varepsilon+2)}{3c} \nabla I,$$

The full force \mathbf{F}_{pm} can be obtained by integrating of \mathbf{f}_{pm} over the nanosheet's volume. The intensity gradient is nonzero only in the region of the edge of the laser spot, in the streak, the area of which we estimated as $a \times a_x$ with $a = 3.7 \mu\text{m}$ and $a_x = 1 \mu\text{m}$. Therefore,

$$|\mathbf{F}_{pm}| = \frac{2(\varepsilon-1)(\varepsilon+2)}{3c} a a_x L \nabla I,$$

where L is the nanosheet's thickness. Using an experimental value $\varepsilon(1040 \text{ nm}) \approx +1.6$ [R7], we get $F_{pm} = 178 \text{ nN}$, which is slightly greater but close to our result obtained using the point-dipole approximation. Since this estimate is more rigorous and does not require the point-dipole approximation, we replace our previous estimate in the revised manuscript with it.

3. The stated theory does not imply any restrictions on the duration of the electromagnetic field, if the intensity I is taken correctly. Unlike CW radiation case, when the average intensity should be used, we substitute the intensity of a laser pulse, thereby taking into account its duration. At that, we do not need to consider an equation of motion of the nanosheet during the pulse, because the estimates for the total force yielding by the pulse indicates its low level unable to drive the nanosheet.

4. Theoretically, a permittivity gradient can arise due to ultrafast refraction nonlinearities like optical Kerr effect or free-carrier absorption. In this case the corresponding terms in Eq. (R1):

$$\mathbf{f}_{pm}^{(\nabla\varepsilon)} = \frac{2\varepsilon_0(\varepsilon-1)}{3} \mathbf{E}^2 \nabla \varepsilon$$

may be important. VSe₂ nanosheets are only known by certain saturable absorption [R8, R9], which gives no impact into the optical force. However, an ultrafast nonlinear refraction index for another 2D semimetal, ZrTe₂, has been reported: $n_2 = 4 \times 10^{-20} \text{ m}^2/\text{W}$ [R10]. At the typical intensity in our experiment of $I = 4 \times 10^{14} \text{ W/m}^2$, it can induce $\Delta n = 1.6 \times 10^{-5}$, which at the length a_x would

result in $\nabla_x \varepsilon = 40.5 \text{ m}^{-1}$. With these estimates, $F_{pm}^{(\nabla \varepsilon)} = \mathbf{f}_{pm}^{(\nabla \varepsilon)} a^2 L = 6 \text{ pN}$, which is negligible even the nonlinearity would turn out several orders of magnitude larger.

In the revised manuscript, we added this analysis as Supplementary Note 6, and used these results for discussion in the “2. Optical force” subsection of the main text.

References

- R1. Kosky, P. *et al.* Exploring Engineering, an Introduction to Engineering and Design, Elsevier, 2021. Ch. 12.
- R2. Borghese, F. *et al.* Optical trapping of nonspherical particles in the T-matrix formalism. *Opt. Express* **15**, 11984 (2007).
- R3. Dienerowitz, M., Mazilu, M., & Dholakia, K. Optical manipulation of nanoparticles: a review. *J. Nanophotonics* **2**, 021875 (2008).
- R4. Donato, M. G. *et al.* Optical trapping and optical force positioning of two-dimensional materials. *Nanoscale* **10**, 1245 (2018).
- R5. Gordon, J. P. Radiation forces and momenta in dielectric media. *Phys. Rev. A* **8**, 14 (1973).
- R6. Tamm, I. E. The fundamentals of the theory of electricity, Mir Publishers, Moscow, 1979. Ch.1 (sec. 1.17) & Ch.2 (sec. 2.13).
- R7. Bayliss, S. C. & Liang, W. Y. Reflectivity and band structure of 1T-VSe₂. *J. Phys. C: Solid State Phys.* **17**, 2193 (1984).
- R8. Li, L. *et al.* Optical pulse modulators based on layered vanadium diselenide nanosheets. *Nanotechnology* **33**, 065203 (2022).
- R9. Li, X. *et al.* High-performance vanadium diselenide nanosheets for the realization of compact pulsed fiber lasers. *Ann. Phys.* **533**, 2100230 (2021).
- R10. Maldonado, M. *et al.* Femtosecond nonlinear refraction of 2D semi-metallic redox exfoliated ZrTe₂ at 800 nm. *Appl. Phys. Lett.* **118**, 011101 (2021).

List of changes in the main text of the revised manuscript:

1. In Author part:

“Yan Wang” are added as one of the contributing authors.

2. In Abstract part:

The words “tens and hundreds of megapascals of surface density” are added.

The sentence “We found that the pulsed irradiation mode is important for this optical drive mechanism, which we consider as photoacoustic.” is added.

3. In Introduction part:

The sentence “The optical force, originating from momentum conservation, is usually small and mainly utilized to transport transparent particles,” is changed to “The optical force, originating from momentum conservation and ponderomotive action of light, is usually small and mainly utilized to transport three-dimensional (3D), or suspended two-dimensional (2D) particles.”.

4. In Results part:

The sentence “Then moving the light source in any direction causes the nanosheet to be driven in the same direction.” is changed to “Then moving the light source in the direction towards the nanosheet provides the uniformity of its motion.”.

The word “a quasi-continuous laser” is changed to “a high repetition frequency laser”.

5. In Discussions part:

In “1. Van der Waals adhesion and friction forces” subsection, the sentences “To test our calculations, we cut one of the nanosheets together with the sapphire substrate by a focused ion beam and examined the contact area with a high-resolution transmission electron microscopy (HRTEM), as shown in Fig. 3 (See details in Supplementary Note 4). Figure 3d indicates the interlayer distance of 0.224 ± 0.034 nm in the central part of the nanosheet. It is in agreement with similar observation of the VSe₂-sapphire contact, and the value by hard-sphere model corresponds to these both observations. Based on the latter value and using the above A value, the surface density of the vdW force between the VSe₂ nanosheet and sapphire substrate (“vdW pressure”) calculated by Eq. (1) holds $F_{vdW}/S = 0.200$ GPa.

It can be clearly seen in Figs. 3b and 3c that air gaps exist along the edges of the nanosheet. Figures 3e and 3f show the enlarged images of air gaps, which are obviously caused by fractures of the layered material during its coating, so the area of close contact is always less than or equal to the visible area of the nanosheet. Therefore, the absolute value of the vdW force, determined by Eq. (1) for a specific nanosheet, is its upper estimate.” are added.

In “1. Van der Waals adhesion and friction forces” subsection, Fig. 3 is added.

In “2. Optical force” subsection, the second paragraph is changed to “The transversal gradient forces are ponderomotive,³⁵ and in our case originates from the gradient of the light intensity, ∇I , in the surface of the nanosheet: $F_{grad} = 2V(\epsilon - 1)(\epsilon + 2)\nabla I/3c$ (See details in Supplementary Note 6). Directed towards the increment of the intensity gradient, this force may influence the lateral deviation of the particle and fix it at the center of the light focus. In our experiment, the laser was reshaped from a Gaussian beam into a near-flat beam via an apodizing filter. Thus the radial gradient was substantially reduced, except the edge intensity gradients, which did not surpass 1×10^{21} W/m³ (assuming the width of the spot boundary of the order of wavelength: $dx \sim 1 \mu\text{m}$). In a nanosheet with a characteristic length $a = 3.7 \mu\text{m}$, $\nabla I \neq 0$ only within a streak of the width $a_x = 1 \mu\text{m}$. Therefore, the volume affected by the force is $V = aa_xL$. With an experimental value of permittivity of VSe₂: $\epsilon(1040 \text{ nm}) = 1.6$, we get the assessment $F_{grad} = 178 \text{ nN}$. This small value, as well as the observed behavior of the nanosheet, on which the force acts mainly in the center of the light spot and weakens towards the periphery, allow us to conclude that optical forces do not participate in the observed effect.”

In “3.1 Nanosheet thermal analysis” subsection, the formulas “ $\Delta T_{av}(t) = L^{-1} \int_0^L \Delta T(z, t) dz$ ” and “ $\Delta T_{int}(t) = T(0, t) - T(L, t)$ ” are added.

In “3.1 Nanosheet thermal analysis” subsection, the sentence “Such a situation may occur when the nanosheet is in island-like contact with the substrate due to accidental defects, such as the presence of dust nanoparticles or, more likely, surface fractures that can lift the nanosheet from the edges.” is changed to “As it is demonstrated by the cross-sectional TEM images in Fig. 3 and Supplementary Note 4, such a situation occurs at the edges of a ME nanosheet, where these gaps are apparently formed during exfoliation due to the separation of the nanosheet from the bulk material with uneven delamination of the layers along the edges.”.

In “3.2 Thermoelastic effect” subsection, the last paragraph is changed to “However, it already helps to understand some of the observed peculiarities at a semi-quantitative level. Thus, the efficiency of movement on different substrates can be considered from the point-of-view of their different thermal conductivities (See details in Supplementary Note 10). The analysis shows that in the case of the quartz glass substrate which possess a lower thermal diffusivity, the temperature increment is remarkably higher in the parts of the nanosheet which closely contact the substrate, but not in the free-standing edges. This leads to a lower asymmetry of the in-plane thermal stress and of the corresponding thermoelastic wave momentum. On the other hand, the friction force is weaker, that leads to large t_{end} , which depends on σ_{fr} nonlinearly. According to our estimates, the possible value for the VSe₂-Quartz contact is $t_{end} \approx 255$ ns. The momentum oscillation asymmetry at $t \gg 1$ ns provides the constant speed of $v_{NS} = 2.8$ m/s, which results in the nanosheet’s displacement of $\Delta x \approx 0.7$ nm. This explains a better efficiency of movement on quartz glass substrate than on sapphire substrate. The immobility on silicon substrate is harder due to the lower value of thermal diffusivity, but rather relates to the increased friction: t_{end} shortens nonlinearly with even small increases in σ_{fr} , and Δx becomes insignificant.

The thermoelastic effect described here also explains the importance of the pulse radiation regime for the optical driving. For the case of a CW radiation or a high-repetition radiation with the heat accumulation regime, the heating is slow compared to the establishment of thermal equilibrium along the nanosheet. Indeed, for the characteristic time of the latter, τ_{TR}^{\parallel} , the maximal possible heating of the free-standing edges is $I\tau_{TR}^{\parallel}(1 - R)(1 - e^{-\alpha L})/\rho C_p L = 6.5$ K, which is not enough to produce the optical driving effect. In the subsequent time, thermal equilibrium is established in the nanosheet, and no asymmetry in the thermal stress and the related elastic wave momentum is appeared.”.

In “3.3 Photophoretic force” subsection, the sentence “However, a simple evaluation,⁸ $F_{PP} \approx -1.4 \times 10^{-6} \times S \times dT/dx \leq 10^{-5}$ N, indicates insignificance of mechanisms of this kind in our system.” is changed to “However, upon a simple evaluation,⁹ the surface density of photophoretic forces: $F_{PP}/S \approx -4.3 \times 10^{-6} dT/dx = 6.4$ kPa (at a temperature gradient $dT/dx = -1.5 \times 10^9$ K/m corresponding to Fig. 5a), indicates insignificance of mechanisms of this kind in our system.”.

6. In References part:

The references “*Nanoscale* 10, 1245 (2018)”, “*Phys. Rev. A* 8, 14 (1973)”, and “*J. Phys. C: Solid State Phys.* 17, 2193 (1984)” are added.

REVIEWERS' COMMENTS

Reviewer #2 (Remarks to the Author):

I appreciate the experimental observation on the flake-substrate gap, which makes the explanation more convincing. The manuscript is now in a better shape and may be accepted for publication.

Reviewer #3 (Remarks to the Author):

The additional experiments and discussions are helpful. Just a few more comments: Note that when the light is focused by an objective, it does not "fall perpendicular to its surface". And scattering forces seldom lead to optical trapping except in two counter-propagating laser beams or when the trapping wavelength is on the blue side of the plasmon resonance. However, scattering forces can lead to transverse motion of an object. This happens when there is a phase gradient of light on the object, such as the nanosheet here. I noticed that the authors used an apodizing filter to shape the Gaussian beam. If the beam center is not exactly aligned with the filter center, phase gradient can potentially arise and create stronger forces (compared to gradient forces) to move the nanosheet. The authors may find some examples on optical manipulation with phase gradient force in literature and consider this possibility.

Responses to Reviewers

Reviewer #3 (Remarks to the Author):

The additional experiments and discussions are helpful. Just a few more comments: Note that when the light is focused by an objective, it does not “fall perpendicular to its surface”. And scattering forces seldom lead to optical trapping except in two counter-propagating laser beams or when the trapping wavelength is on the blue side of the plasmon resonance. However, scattering forces can lead to transverse motion of an object. This happens when there is a phase gradient of light on the object, such as the nanosheet here. I noticed that the authors used an apodizing filter to shape the Gaussian beam. If the beam center is not exactly aligned with the filter center, phase gradient can potentially arise and create stronger forces (compared to gradient forces) to move the nanosheet. The authors may find some examples on optical manipulation with phase gradient force in literature and consider this possibility.

Reply: We thank the reviewer for drawing attention to such a curious phenomenon as the phase gradient.

We have to note that we observed many times during our experiments, and now we checked it again, that centering the apodizing filter relative to the optical axis, which was sharply monitored by the microscope camera, does not affect the character of the nanosheet’ motion. To check it up, we decentered the filter by shifting it to different directions perpendicular to the beam and observed no speed nor direction changes in the nanosheet displacement. Moreover, the nanosheets also moves under Gaussian beam without any apodization. According to the well-known field profile in the Gaussian beam:

$$E(r, z) = E_0 \frac{w_0}{w(z)} \exp\left(-\frac{r^2}{w(z)^2}\right) \exp\left[-ik\left(z + \frac{r^2}{2R(z)}\right)\right],$$

the radial part of the phase gradient is inverse to the wavefront curvature radius $R(z)$ and thus should turn to zero in the focal plane of the microscope. This outcome further

confirms the fact that the phase gradient effect can be negligible. In our experiment, we have used the apodization filter in order to reduce intensity gradients that looks more valuable for the transversal forces, and therefore we needed this excuse.

In the revised manuscript, we added appropriate explanations in “Optical forces” subsection.

List of changes in the main text of the revised manuscript:

1. In “Optical forces” subsection:

The paragraph “The transversal forces exerting on a nanoparticle from light come from phase and intensity gradients. Despite the piconewton values of the gradient forces, their application for sorting metallic nanoparticles has been reported. The phase gradient in the focus of a Gaussian beam is zero, and an appropriate phase mask is necessary to generate the forces. Although the apodizing filter that we use in our experiment is radially symmetric, alignment errors can bring to the appearance of some phase gradient. However, our careful observations of the nanosheet movement with the transversal shifting of the filter indicate the absence of the influence of these forces.” is added.

2. In References section:

The references “*Phys. Rev. Lett.* 100, 013602 (2008)” and “*Nano Lett.* 18, 4500 (2018)” are added.